# Exportin Crm1 is repurposed as a docking protein to generate microtubule organizing centers at the nuclear pore

Xun X Bao[1], Christos Spanos[1], Tomoko Kojidani[2,3], Eric M Lynch[1†], Juri Rappsilber[1,4], Yasushi Hiraoka[2,5], Tokuko Haraguchi[2,5], Kenneth E Sawin[1*]

[1]Wellcome Centre for Cell Biology, School of Biological Sciences, University of Edinburgh, Edinburgh, United Kingdom; [2]Advanced ICT Research Institute Kobe, National Institute of Information and Communications Technology, Kobe, Japan; [3]Department of Chemical and Biological Sciences, Faculty of Science, Japan Women's University, Tokyo, Japan; [4]Department of Bioanalytics, Institute of Biotechnology, Technische Universität Berlin, Berlin, Germany; [5]Graduate School of Frontier Biosciences, Osaka University, Suita, Japan

**Abstract** Non-centrosomal microtubule organizing centers (MTOCs) are important for microtubule organization in many cell types. In fission yeast *Schizosaccharomyces pombe*, the protein Mto1, together with partner protein Mto2 (Mto1/2 complex), recruits the γ-tubulin complex to multiple non-centrosomal MTOCs, including the nuclear envelope (NE). Here, we develop a comparative-interactome mass spectrometry approach to determine how Mto1 localizes to the NE. Surprisingly, we find that Mto1, a constitutively cytoplasmic protein, docks at nuclear pore complexes (NPCs), via interaction with exportin Crm1 and cytoplasmic FG-nucleoporin Nup146. Although Mto1 is not a nuclear export cargo, it binds Crm1 via a nuclear export signal-like sequence, and docking requires both Ran in the GTP-bound state and Nup146 FG repeats. In addition to determining the mechanism of MTOC formation at the NE, our results reveal a novel role for Crm1 and the nuclear export machinery in the stable docking of a cytoplasmic protein complex at NPCs.
DOI: https://doi.org/10.7554/eLife.33465.001

**\*For correspondence:**
ken.sawin@ed.ac.uk

**Present address:** †Department of Biochemistry, University of Washington, Seattle, United States

**Competing interests:** The authors declare that no competing interests exist.

## Introduction

Non-centrosomal microtubule organizing centers (MTOCs) are critical to the morphology and function of many types of cells (*Petry and Vale, 2015*; *Sanchez and Feldman, 2017*; *Wu and Akhmanova, 2017*), especially cells in which interphase microtubules (MTs) are arranged in linear rather than radial arrays (*Bartolini and Gundersen, 2006*). Examples include differentiated animal cells such as neurons (*Kapitein and Hoogenraad, 2015*), muscle (*Mogessie et al., 2015*; *Tassin et al., 1985*), and epithelial cells (*Wu and Akhmanova, 2017*), and many higher plant cells (*Masoud et al., 2013*; *Oda, 2015*), as well as some single-celled eukaryotes, such as fission yeast *Schizosaccharomyces pombe* (*Chang and Martin, 2009*; *Sawin and Tran, 2006*).

The mechanisms underlying non-centrosomal MTOC formation are just beginning to be understood. Some non-centrosomal MTs are thought to be generated by nucleation-and-release from the centrosome, followed by minus-end stabilization and anchoring elsewhere in the cell (*Bartolini and Gundersen, 2006*; *Sanchez and Feldman, 2017*; *Wu and Akhmanova, 2017*). However, in many cases, MTs are nucleated directly from non-centrosomal sites by the γ-tubulin complex, the primary microtubule-nucleation complex in eukaryotic cells (*Kollman et al., 2011*; *Petry and Vale, 2015*).

**eLife digest** Nearly all cells contain networks of filaments called microtubules that play many different roles. They provide internal structure; they serve as 'tracks' for transporting materials from one region of the cell to another; and they help to separate chromosomes during cell division. To understand how microtubules work, it is important to know how they are organized and distributed in cells.

In several types of cells, including muscle cells in humans, and most plant cells, microtubules form on the surface of the cell nucleus – the membrane-bound compartment that stores genetic information. A key protein involved in microtubule formation, called Mto1, is present on the surface of the nucleus, but it was not clear how Mto1 localizes there.

Using fission yeast cells, Bao et al. have devised a new method to identify the proteins that recruit Mto1 to the surface of the nucleus. This revealed that Mto1 is recruited to the nuclear pores – large channels on the surface of the nucleus through which proteins can be transported. Unexpectedly, the proteins that recruit Mto1 to the nuclear pores do so using a mechanism that is also used to transport proteins out of the nucleus. Thus, in addition to determining how microtubules are organized at the cell nucleus, Bao et al. have identified a previously unknown role for the 'nuclear transport machinery'.

Currently, drugs that inhibit the nuclear transport machinery form potential treatments for some cancers and viral infections. A better understanding of the multiple roles performed by the nuclear transport machinery may help researchers to design more effective inhibitor drugs.
DOI: https://doi.org/10.7554/eLife.33465.002

Understanding how the γ-tubulin complex is recruited to these sites is thus key to deciphering the fundamental mechanisms of non-centrosomal MT organization (*Lin et al., 2015*).

Sites of non-centrosomal γ-tubulin complex recruitment include pre-existing microtubules themselves, as well as membrane-bound compartments such as the Golgi apparatus and the nuclear envelope (NE). Recruitment of the γ-tubulin complex to pre-existing microtubules depends on the multi-subunit augmin complex, in both animals and plants (*Goshima et al., 2008*; *Liu et al., 2014*; *Sánchez-Huertas et al., 2016*). Microtubule nucleation and organization by the Golgi apparatus is orchestrated largely by AKAP450, which recruits not only the γ-tubulin complex but also its activators, as well as MT minus-end stabilizers (*Rivero et al., 2009*; *Wu et al., 2016*). Combined recruitment of γ-tubulin complex and MT minus-end stabilizers/anchoring proteins is also important for MTOC organization at the cell cortex in diverse types of epithelial cells (summarized in [*Sanchez and Feldman, 2017*; *Wu and Akhmanova, 2017*]).

MTOC formation at the NE remains poorly understood. The NE is an important MT nucleation site both in muscle cells (*Tassin et al., 1985*) and in higher plants (*Ambrose and Wasteneys, 2014*; *Masoud et al., 2013*; *Stoppin et al., 1994*), as well as in fission yeast (*Lynch et al., 2014*; *Sawin and Tran, 2006*). In muscle, γ-tubulin complex components and associated proteins are redistributed from the centrosome to the NE during development/differentiation, coincident with a decrease in centrosomal MT nucleation and large-scale changes in intracellular MT organization (*Bugnard et al., 2005*; *Fant et al., 2009*; *Srsen et al., 2009*; *Zebrowski et al., 2015*). In plant cells, which lack centrosomes altogether, many of the same proteins are similarly observed on the NE, especially before and/or after cell division (*Erhardt et al., 2002*; *Janski et al., 2012*; *Nakamura et al., 2012*; *Seltzer et al., 2007*). However, the mechanisms that regulate their recruitment are largely a mystery.

Fission yeast nucleate MTs from multiple non-centrosomal sites through the cell cycle and thus provide an excellent system to study non-centrosomal MTOCs, including those on the NE (*Sawin and Tran, 2006*). During interphase, linear arrays of MTs are nucleated from the spindle pole body (SPB; the yeast centrosome equivalent), from MTOCs on the NE and on pre-existing microtubules, and from 'free' MTOCs in the cytoplasm. As cells enter mitosis, non-centrosomal MT nucleation is switched off (*Borek et al., 2015*) and the duplicated SPBs become the only active MTOCs, nucleating both intranuclear spindle MTs and cytoplasmic astral MTs. Toward the end of cell division,

microtubules are nucleated from the cytokinetic actomyosin ring (CAR). By contrast, in budding yeast *Saccharomyces cerevisiae*, the SPBs are the only MTOCs throughout the cell cycle.

In fission yeast, all types of MT nucleation in the cytoplasm (i.e. both centrosomal and non-centrosomal nucleation) depend on the Mto1/2 complex (*Janson et al., 2005*; *Samejima et al., 2005*; *Sawin et al., 2004*; *Venkatram et al., 2005*; *Venkatram et al., 2004*). Mto1/2 contains multiple copies of the proteins Mto1 and Mto2 and directly recruits the γ-tubulin complex to prospective MTOC sites. Mto1/2 interacts with the γ-tubulin complex via Mto1's Centrosomin Motif 1 (CM1) domain, which is conserved in higher eukaryotic MTOC regulators such as *Drosophila* centrosomin, and human CDK5RAP2 and myomegalin (*Samejima et al., 2008*; *Sawin et al., 2004*; *Zhang and Megraw, 2007*). Interaction of CM1-domain proteins with the γ-tubulin complex can also serve to activate the γ-tubulin complex (*Choi et al., 2010*; *Lynch et al., 2014*), although the detailed mechanisms remain unclear.

Because Mto1/2 localizes to prospective MTOC sites independently of interacting with the γ-tubulin complex (*Samejima et al., 2008*), Mto1/2 localization effectively determines where and when all cytoplasmic MTOCs are generated, and thus understanding Mto1/2 localization is critical to understanding MTOC formation more broadly. Mto1/2 localization is mediated primarily by domains within Mto1 (*Figure 1A*; [*Samejima et al., 2010*]), although Mto2 contributes indirectly by helping to multimerize the Mto1/2 complex (*Lynch et al., 2014*; *Samejima et al., 2005*). Mto1/2 association with pre-existing MTs depends on a broadly defined region near the Mto1 C-terminus, while localization to the CAR and the SPB is mediated by overlapping modular sequences within the conserved MASC domain at the Mto1 C-terminus (*Samejima et al., 2010*). Localization to the CAR involves interaction of MASC with the unconventional myosin Myp2, while localization to the SPB involves the Septation Initiation Network protein Cdc11 (*Samejima et al., 2010*).

Here, we determine the mechanism of Mto1/2 localization to the NE. Using a comparative-interactome mass spectrometry approach, we find that NE localization depends on the Mto1 N-terminus interacting with exportin Crm1, a nuclear transport receptor, and nucleoporin Nup146, a component of the nuclear pore complex (NPC). We further find that although Mto1 is an exclusively cytoplasmic protein, it becomes stably docked at the NPC by mimicking a nuclear export cargo. In addition to revealing the mechanism of MTOC formation at the fission yeast NE, our work demonstrates a completely novel role for the nuclear export machinery, in which the exportin is repurposed to create NPC-docking sites for cytoplasmic proteins with functions unrelated to nuclear export.

## Results

### MT nucleation from the NE contributes to nuclear positioning

Mto1 localization to the NE is enhanced in the C-terminal truncation mutant Mto1[NE], which lacks MASC and MT-localization domains ([*Lynch et al., 2014*]; *Figure 1A*). Previously, we deleted amino acids 1–130 from Mto1[NE] and from full-length Mto1 to make Mto1[bonsai] and Mto1[Δ130], respectively (*Figure 1A*), and we showed that these deletions lead to loss of Mto1/2 complex from the NE, accompanied by loss of MT nucleation from the NE (*Lynch et al., 2014*). However, in that work the consequences of this altered MT nucleation were not investigated. In fission yeast, MT-dependent pushing forces are thought to center the interphase nucleus precisely in the cell middle (*Tran et al., 2001*). Because nuclear position during early mitosis determines the future cell division plane, this ensures equal size of daughter cells after cell division (*Daga and Chang, 2005*). To investigate whether MT nucleation from the NE contributes to nuclear positioning, we measured interphase nuclear position in *mto1-GFP*, *mto1[NE]-GFP*, *mto1[Δ130-GFP]* and *mto1[bonsai]-GFP* cells (*Figure 1A*; *Figure 1—figure supplement 1*). (In these and all subsequent experiments, *mto1* mutants replace endogenous wild-type *mto1+* at the *mto1* locus, and in this particular experiment, all versions of *mto1* were GFP-tagged to equalize protein expression levels [*Lynch et al., 2014*]). Interestingly, nuclear positioning was less accurate in *mto1[bonsai]-GFP* and *mto1[Δ130]-GFP* cells compared to *mto1[NE]-GFP* and *mto1-GFP* cells, indicating that MT nucleation from the NE contributes to nuclear positioning. By contrast, there was no difference in nuclear positioning between wild-type and *mto1[NE] cells*, or between *mto1[131–1115]* and *mto1[bonsai]* cells, indicating that MT nucleation from the SPB is not particularly important for nuclear positioning.

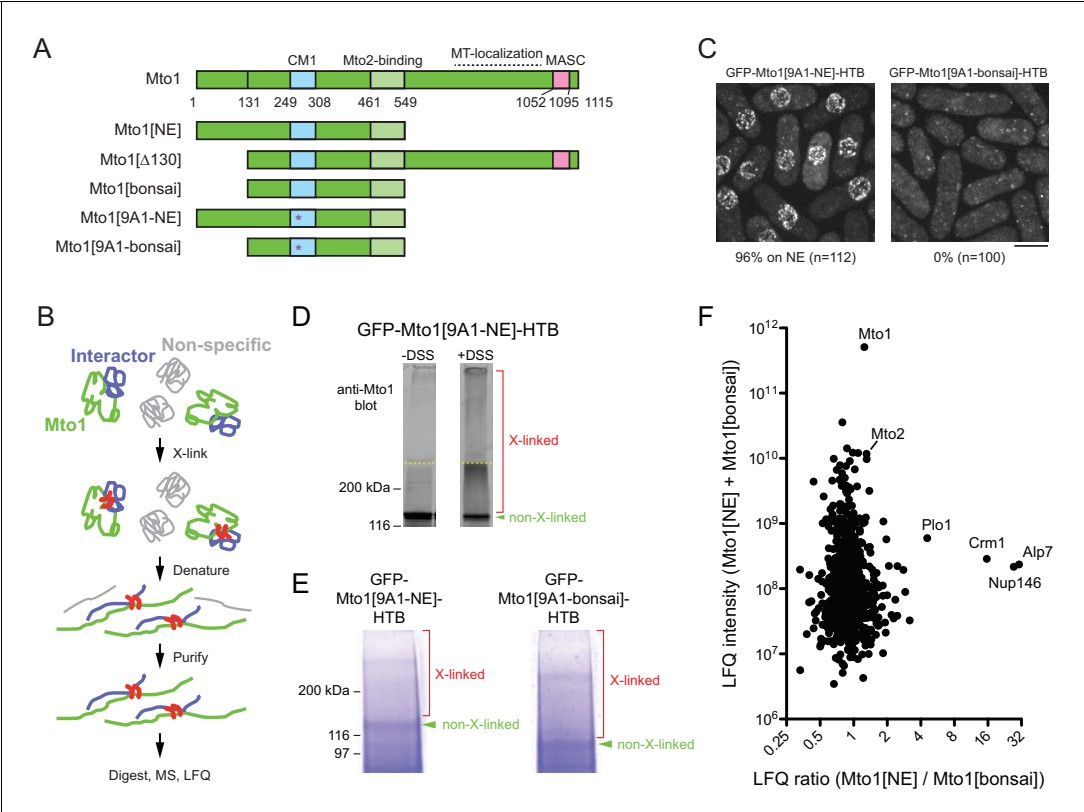

**Figure 1.** Identification of proteins interacting with Mto1[NE] but not Mto[bonsai]. (**A**) Diagram of full-length Mto1 and Mto1-truncation mutants. Asterisk indicates *9A1* mutation, which abolishes interaction with γ-tubulin complex (*Samejima et al., 2008*). (**B**) Outline of cross-linking and mass spectrometry approach to identify Mto1 interactors. (**C**) Localization of GFP-Mto1[9A1-NE]-HTB and GFP-Mto1[9A1-bonsai]-HTB. Numbers below images indicate percent cells with GFP signal on the nuclear envelope (n = total number of cells scored). (**D**) Anti-Mto1 Western blot of whole-cell lysates from *GFP-mto1[9A1-NE]-HTB* cells in the absence of cross-linking (-DSS) and after disuccinimidyl suberate cross-linking (+DSS). Dashed line indicates boundary between resolving gel and stacking gel. (**E**) SDS-PAGE of GFP-Mto1[9A1-NE]-HTB sample and GFP-Mto1[9A1-bonsai]-HTB sample after DSS cross-linking and two-step denaturing purification. Regions marked 'X-linked' were analyzed by mass spectrometry (see Materials and methods). (**F**) Mass spectrometry label-free quantification (LFQ) of 750 proteins from samples as in E. 'LFQ ratio' indicates relative enrichment of a given protein in the purified GFP-Mto1[9A1-NE]-HTB sample compared to the purified GFP-Mto1[9A1-bonsai]-HTB sample. 'LFQ intensity' indicates total intensity (arbitrary units) of a given protein from the combined purified samples. Data shown represent one of two independent biological replicates. See also *Table 1*. Complete datasets are in *Supplementary file 3*. Bar, 5 µm.

DOI: https://doi.org/10.7554/eLife.33465.003

The following figure supplements are available for figure 1:

**Figure supplement 1.** Nuclear positioning in *mto1* mutants.
DOI: https://doi.org/10.7554/eLife.33465.004

**Figure supplement 2.** The *mto1 9A1* mutation enhances Mto1[NE]-GFP localization to the nuclear envelope.
DOI: https://doi.org/10.7554/eLife.33465.005

**Figure supplement 3.** Alp7 is not required for Mto1[NE] localization to the nuclear envelope.
DOI: https://doi.org/10.7554/eLife.33465.006

## Identification of proteins interacting with Mto1[NE] but not with Mto1[bonsai]

To identify proteins involved in recruiting Mto1 to the NE, we wanted to compare interactomes of Mto1[NE] vs. Mto1[bonsai]. Initially, we attempted to use SILAC mass spectrometry (MS) (*Bicho et al., 2010*; *Ong et al., 2002*) to compare anti-GFP immunoprecipitates of Mto1[9A1-NE]-GFP and Mto1[9A1-bonsai]-GFP, which are otherwise identical to Mto1[NE]-GFP and Mto1[bonsai]-GFP except for the additional mutation of nine consecutive amino acids in the CM1 domain to alanine (*Samejima et al., 2008*), *Figure 1A*); the *9A1* mutation disrupts interaction with the γ-tubulin complex and thereby enhances localization of Mto1[NE] to the NE ([*Lynch et al., 2014*]; *Figure 1—*

*figure supplement 2*). In preliminary experiments, however, we found that the immunoprecipitation approach yielded low peptide counts for many Mto1-interactors of potential interest (*Supplementary file 2*). We therefore decided to develop a more robust method to capture interactors even when they may be low-abundance and/or low-affinity interactors.

We tagged Mto1[9A1-NE] and Mto1[9A1-bonsai] at their N-termini with GFP and at their C-termini with an HTB (His-TEV-biotin) tag, which allows for two-step purification of a tagged protein under fully denaturing conditions after cross-linking to interactors (*Tagwerker et al., 2006*) (*Figure 1B*). As expected, GFP-Mto1[9A1-NE]-HTB localized to the NE in vivo, while GFP-Mto1[9A1-bonsai]-HTB was present only in the cytoplasm (*Figure 1C*). Disuccinimidyl suberate (DSS) cross-linking of cell cryogrindates shifted a significant proportion of HTB-tagged Mto1 into higher molecular-weight species (*Figure 1D*). After DSS cross-linking and denaturing purification (*Figure 1E*; see Materials and methods), we analyzed cross-linked adducts of GFP-Mto1[9A1-NE]-HTB and GFP-Mto1[9A1-bonsai]-HTB by label-free quantification (LFQ) MS ([*Cox and Mann, 2008*; *Tyanova et al., 2016*]; *Figure 1F*; *Table 1*; *Supplementary file 3*). Among the proteins significantly enriched in the Mto1[9A1-NE] interactome vs. the Mto1[9A1-bonsai] interactome, we identified nucleoporin Nup146 (*Asakawa et al., 2014*; *Chen et al., 2004*), exportin Crm1 (*Adachi and Yanagida, 1989*; *Fung and Chook, 2014*; *Hutten and Kehlenbach, 2007*; *Stade et al., 1997*), the fission yeast TACC homolog, Alp7 (*Sato et al., 2004*), and, to a lesser extent, polo kinase Plo1 (*Ohkura et al., 1995*).

Neither Alp7 nor Plo1 is known to localize to the NE, and Plo1 was not investigated further. The interaction of Mto1[NE] with Alp7 was of potential interest because of the role of Alp7 in microtubule organization (*Ling et al., 2009*; *Sato et al., 2009*; *Zheng et al., 2006*), and an interaction between Mto1 and Alp7 has been confirmed independently (M. Sato, Waseda University, personal communication, July 2017). However, we found that in *alp7Δ* deletion mutants, Mto1[9A1-NE]-GFP was present on the NE just as in wild-type (*alp7+*) cells (*Figure 1—figure supplement 3*). This indicates that Alp7 is not required for Mto1 localization to the NE.

## Mto1[NE] associates with the cytoplasmic face of the NPC

The interaction of Mto1[9A1-NE] with Nup146 suggested that Mto1 may localize to nuclear pore complexes (NPCs) on the NE. We therefore imaged Mto1[9A1-NE]-GFP together with Nup146-3mCherry in a *nup132Δ* background, in which NPCs can become clustered on the NE (*BaiBaï et al., 2004*). We observed extensive colocalization of Mto1[9A1-NE]-GFP with Nup146-3mCherry clusters (*Figure 2A*), indicating specific association with NPCs.

**Table 1.** Data for selected proteins from mass spectrometry comparison of GFP-Mto1[9A1-NE]-HTB and GFP-Mto1[9A1-bonsai]-HTB interactomes.

Peptide counts and label-free quantification (LFQ) values for selected proteins shown in *Figure 1F*. Data from two independent biological replicates are shown. Nsp1 and Nup82 are included as likely representative Nup146 interactors, based on homology to budding yeast (*Belgareh et al., 1998*). See also *Supplementary file 3*.

| Protein | Replicate 1 (E160307) | | | | | Replicate 2 (E161126) | | | | |
|---|---|---|---|---|---|---|---|---|---|---|
| | Peptides from strain KS7611 | Peptides from strain KS8371 | LFQ intensity from strain KS7611 | LFQ intensity from strain KS8371 | LFQ ratio | Peptides from strain KS7611 | Peptides from strain KS8371 | LFQ intensity from strain KS7611 | LFQ intensity from strain KS8371 | LFQ ratio |
| Alp7 | 17 | 3 | 2.3e8 | 7.5e6 | 30.5 | 14 | 5 | 1.9e8 | 3.7e7 | 5.2 |
| Crm1 | 20 | 5 | 2.7e8 | 1.7e7 | 15.6 | 18 | 5 | 2.7e8 | 2.0e7 | 13.7 |
| Mto1 | 59 | 46 | 2.8e11 | 2.3e11 | 1.2 | 52 | 43 | 2.8e11 | 2.5e11 | 1.1 |
| Mto2 | 14 | 20 | 6.6e9 | 5.1e9 | 1.3 | 16 | 22 | 6.7e9 | 5.8e9 | 1.2 |
| Nsp1 | 14 | 11 | 2.5e8 | 1.8e8 | 1.4 | 14 | 12 | 2.2e8 | 1.3e8 | 1.7 |
| Nup146 | 20 | 2 | 2.1e8 | 7.7e6 | 27.3 | 20 | 1 | 2.1e8 | NQ | NQ |
| Nup82 | 13 | 9 | 1.5e8 | 7.3e7 | 2.1 | 9 | 4 | 7.4e7 | 2.2e7 | 3.4 |
| Plo1 | 28 | 13 | 4.9e8 | 1.1e8 | 4.6 | 20 | 9 | 3.3e8 | 6.3e7 | 5.3 |

NQ = not quantified, because peptide count in the relevant sample was below threshold for quantification.
DOI: https://doi.org/10.7554/eLife.33465.007

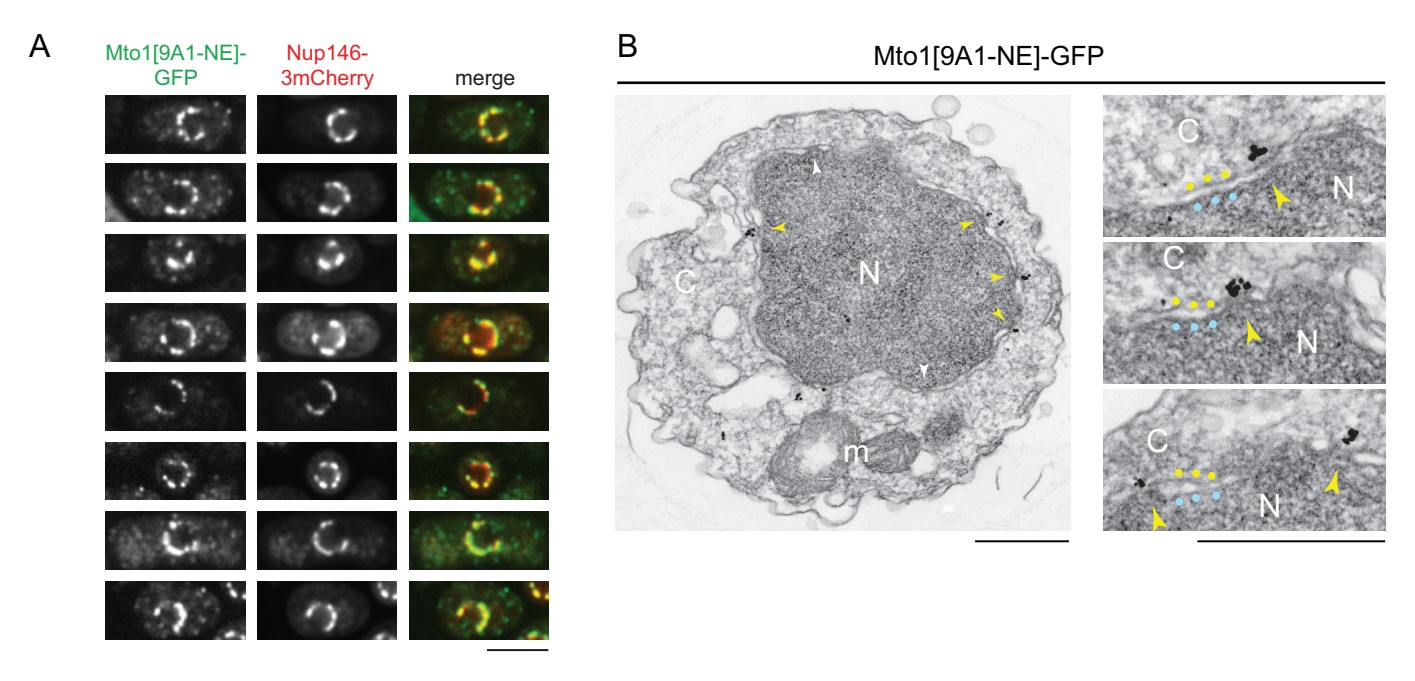

**Figure 2.** Mto1[NE] is localized to the cytoplasmic face of the nuclear pore complex. (**A**) Colocalization of Mto1[9A1-NE]-GFP and Nup146-3mCherry after nuclear pore complex (NPC) clustering in *nup132Δ* cells. For each cell, a single central Z-section is shown. (**B**) Immunoelectron microscopy of Mto1 [9A1-NE]-GFP. Left panel shows cross-section of a single cell. NPCs can be seen as slightly electron-dense regions where inner and outer nuclear membranes meet. Yellow arrowheads indicate NPCs with Mto1[9A1-NE]-GFP on cytoplasmic face of NPC. White arrowheads indicate examples of unstained NPCs. Right panels show magnified examples from other cells. Blue and yellow dots indicate inner and outer nuclear membranes, respectively. N, nucleus; C, cytoplasm; m, mitochondria. Bars, 5 μm (**A**), 0.5 μm (**B**).
DOI: https://doi.org/10.7554/eLife.33465.008

We also examined Mto1[9A1-NE]-GFP localization by immunoelectron microscopy. Close homologs of Nup146 in budding yeast (Nup159; referred to here as *Sc* Nup159) and humans (Nup214; referred to as *Hs* Nup214) are both located exclusively at the cytoplasmic face of NPCs (*Gorsch et al., 1995*; *Kraemer et al., 1994*; *Kraemer et al., 1995*), and indirect evidence suggests that this is also the case for Nup146 (*Lo Presti et al., 2012*). Consistent with this, we observed Mto1 [9A1-NE]-GFP specifically at the cytoplasmic face of NPCs (*Figure 2B*).

## Mto1 localization to NPCs requires export cargo-binding activity of exportin Crm1

The interaction of Mto1[NE] with Crm1 was both surprising and puzzling. As the major transport receptor for nuclear export of proteins (as well as some RNAs), Crm1 normally forms a trimeric complex with export cargo and RanGTP within the nucleus, which facilitates transit of cargo through the permeability barrier of the NPC and into the cytoplasm (*Dong et al., 2009*; *Fung and Chook, 2014*; *Hutten and Kehlenbach, 2007*). However, to date, there is no evidence that Mto1 is a nuclear export cargo or indeed is ever present in the nucleus.

Because deletion of *crm1+* is lethal (*Adachi and Yanagida, 1989*), we investigated the significance of the Mto1-Crm1 interaction by asking whether inhibition of Crm1 cargo-binding activity affects Mto1 localization to NPCs. Nuclear export cargos typically bind to Crm1 via hydrophobic nuclear export signals (NESs) (*Dong et al., 2009*; *Fung and Chook, 2014*; *Fung et al., 2015*; *Güttler et al., 2010*; *Hutten and Kehlenbach, 2007*; *Kutay and Güttinger, 2005*). This can be inhibited by the drug leptomycin B (LMB), which binds within the hydrophobic NES-binding cleft of Crm1 (*Dong et al., 2009*; *Fornerod et al., 1997a*; *Fukuda et al., 1997*; *Fung and Chook, 2014*; *Ossareh-Nazari et al., 1997*). As a result, when cells are treated with LMB, nuclear export cargos accumulate within the nucleus. Interestingly, after LMB treatment, we found that Mto1[9A1-NE]-GFP

was lost from NPCs (*Figure 3A*). Strikingly, however, rather than accumulating in the nucleus, Mto1[9A1-NE]-GFP became dispersed in the cytoplasm.

Given the unusual behavior of Mto1[9A1-NE]-GFP after LMB treatment, we confirmed that LMB was inhibiting nuclear export. We assayed localization of Alp7, which shuttles continuously in and out of the nucleus during interphase, in complex with its partner protein Alp14 (ch-TOG homolog) (*Okada and Sato, 2015*; *Okada et al., 2014*)(*Figure 3—figure supplement 1*). In the absence of LMB, Alp7-3GFP was present in the cytoplasm, primarily as puncta on cytoplasmic MTs. As expected, after LMB treatment, Alp7-3GFP accumulated in the nucleoplasm and on an intranuclear

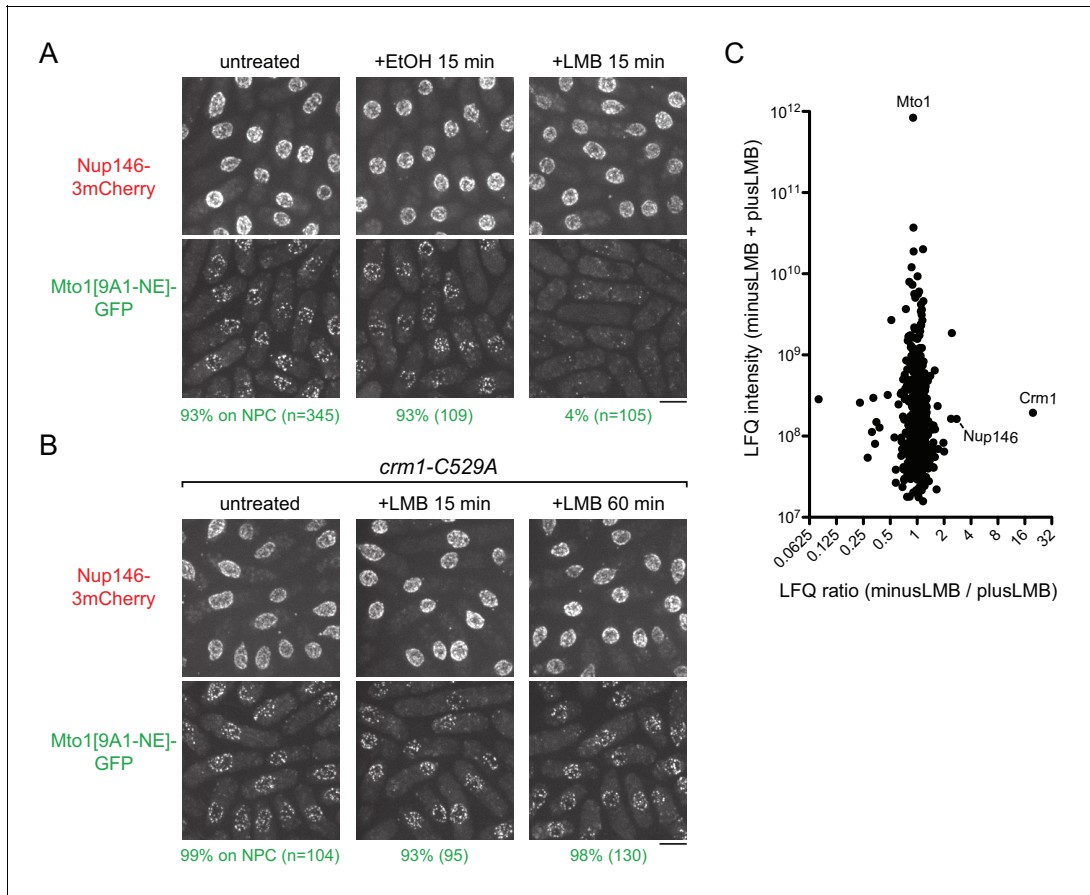

**Figure 3.** Inhibition of Crm1 cargo-binding by Leptomycin B treatment disrupts Mto1[NE] localization to nuclear pore complexes and Mto1[NE] interaction with Crm1. (**A**) Localization of Nup146-3mCherry and Mto1[9A1-NE]-GFP in untreated cells and in cells treated with 1% ethanol (+EtOH) or with 100 nM leptomycin B (+LMB) in ethanol for 15 min. Numbers below images indicate percent cells with Mto1 on NPCs (n = total number of cells scored). (**B**) Localization of Nup146-3mCherry and Mto1[9A1-NE]-GFP in *crm1-C529A* cells, which are resistant to LMB. Cells were untreated or treated with 100 nM LMB for 15 or 60 min. (**C**) Mass spectrometry label-free quantification (LFQ) of 483 proteins from samples of cross-linked, purified GFP-Mto1[9A1-NE]-HTB from untreated ("minusLMB") vs. LMB-treated ("plusLMB") cells. 'LFQ ratio' indicates relative enrichment of a given protein in the purified minusLMB sample compared to the purified plusLMB sample. 'LFQ intensity' indicates total intensity (arbitrary units) of a given protein from the combined purified samples. Data shown represent geometric mean from two independent biological replicates. See also *Table 2*. Complete datasets are in *Supplementary file 4*. Bars, 5 μm.

DOI: https://doi.org/10.7554/eLife.33465.009

The following figure supplements are available for figure 3:

**Figure supplement 1.** Leptomycin B treatment leads to accumulation of Alp7 in the nucleoplasm.
DOI: https://doi.org/10.7554/eLife.33465.010

**Figure supplement 2.** Characterization of *crm1-C529* mutants.
DOI: https://doi.org/10.7554/eLife.33465.011

**Figure supplement 3.** Microtubule regrowth after leptomycin B treatment.
DOI: https://doi.org/10.7554/eLife.33465.012

MT bundle that has been described to form upon LMB treatment of fission yeast (*Matsuyama et al., 2006*)(*Figure 3—figure supplement 1*).

To rule out the possibility that loss of Mto1[9A1-NE]-GFP from NPCs was due to an off-target effect of LMB (i.e. unrelated to Crm1 inhibition), we generated an LMB-resistant *crm1* mutant. LMB is a particularly potent inhibitor of Crm1 because it reacts covalently with cysteine 529 (C529) in Crm1's NES-binding cleft (*Kudo et al., 1999*). We mutated C529 in the endogenous *crm1* coding sequence to alanine (*crm1-C529A*), as well as to serine (*crm1-C529S*), threonine (*crm1-C529T*) and valine (*crm1-C529V*) (*Figure 3B*, *Figure 3—figure supplement 2*). All four mutants were viable, indicating that they preserve essential functions of *crm1* for nuclear export, and three out of the four were resistant to high concentrations of LMB (*Figure 3—figure supplement 2A*). Interestingly, we found that in *crm1-C529A* cells, Mto1[9A1-NE]-GFP localized to NPCs both in the absence and in the presence of LMB (*Figure 3B*). This demonstrates that loss of Mto1 from NPCs after LMB treatment can be specifically attributed to inhibition at the Crm1 cargo-binding cleft.

In addition to these experiments, we used immunofluorescence to compare MT regrowth after cold-induced MT depolymerization in control vs. LMB-treated wild-type cells (i.e. cells expressing full-length, untagged Mto1). Previous work showed that cold-induced MT depolymerization causes the pool of Mto1 normally associated with cytoplasmic MTs to redistribute to the NE, and as a result, when cells are rewarmed, nearly all MT regrowth initiates from the NE (*Sawin et al., 2004*). As expected, we found that in control cells, MT regrowth occurred from the NE. However, in LMB-treated cells, MT regrowth occurred randomly in the cytoplasm, consistent with a failure of Mto1 to localize to the NE after LMB treatment (*Figure 3—figure supplement 3*).

## Mto1 interacts with Crm1 via a nuclear export signal-like sequence

How might Crm1 cargo-binding activity be involved in Mto1 localization to the NPC? We hypothesized that Mto1 itself might bind to Crm1 as an unconventional 'cargo' and somehow exploit this interaction to localize to the cytoplasmic face of the NPC. To test this, we used LFQ MS to compare GFP-Mto1[9A-NE]-HTB interactomes prepared from untreated vs. LMB-treated cells (*Figure 3C*; *Table 2*; *Supplementary file 4*). Interestingly, only 3–4 out of nearly 500 quantified proteins were significantly enriched in the GFP-Mto1[9A1-NE]-HTB interactome from untreated cells compared to LMB-treated cells. Among these, Crm1 showed the greatest enrichment (~20X). Nup146 also showed enrichment, but to a lesser extent (~2.8X), which may indicate that Mto1 can bind weakly to Nup146 independently of Crm1 (see Discussion). These results demonstrate that, like Mto1 localization to NPCs, Mto1 interaction with Crm1 requires Crm1 cargo-binding activity.

**Table 2.** Data for selected proteins from mass spectrometry comparison of GFP-Mto1[9A1-NE]-HTB interactomes from control cells and after treatment with leptomycin B.

Peptide counts and label-free quantification (LFQ) values for selected proteins from the two replicate experiments contributing to the graph in *Figure 3C*. See also *Supplementary file 4*.

| Protein | Replicate 1 (E150924) | | | | | Replicate 2 (E151106) | | | | |
| | Peptides from strain KS7669 -LMB | Peptides from strain KS7669 +LMB | LFQ intensity from strain KS7669 -LMB | LFQ intensity from strain KS7669 + LMB | LFQ ratio | Peptides from strain KS7669-LMB | Peptides from strain KS7669 +LMB | LFQ intensity from strain KS7669 -LMB | LFQ intensity from strain KS7669 + LMB | LFQ ratio |
|---|---|---|---|---|---|---|---|---|---|---|
| Alp7 | 20 | 13 | 2.5e8 | 1.4e8 | 1.7 | 14 | 15 | 6.3e8 | 4.4e8 | 1.4 |
| Crm1 | 17 | 4 | 1.1e8 | 1.3e7 | 9.0 | 19 | 3 | 2.9e8 | 6.8e6 | 43.3 |
| Mto1 | 59 | 58 | 2.6e11 | 2.9e11 | 0.9 | 58 | 59 | 6.1e11 | 6.7e11 | 0.9 |
| Mto2 | 7 | 7 | 2.0e8 | 2.2e8 | 0.9 | 5 | 5 | 6.5e8 | 8.4e8 | 0.8 |
| Nsp1 | 7 | 7 | 6.9e7 | 6.0e7 | 1.2 | 11 | 11 | 3.7e8 | 3.7e8 | 1.0 |
| Nup146 | 14 | 8 | 7.4e7 | 3.3e7 | 2.3 | 12 | 5 | 1.9e8 | 5.6e7 | 3.4 |
| Nup82 | 9 | 10 | 4.5e7 | 7.6e7 | 0.6 | 9 | 9 | 1.7e8 | 2.9e8 | 0.6 |
| Plo1 | 26 | 26 | 2.4e8 | 2.4e8 | 1.0 | 25 | 27 | 8.5e8 | 7.7e8 | 1.1 |

DOI: https://doi.org/10.7554/eLife.33465.013

Based on these findings, we next used the LocNES algorithm (*Xu et al., 2015*) to search for NES-like sequences within the N-terminal 130 amino acids of Mto1, which are present in Mto1[NE] but absent from Mto1[bonsai]. The sequence spanning Mto1 amino acids 9–25 contained two closely overlapping candidate NESs (*Figure 4A*). Interestingly, the spacing of hydrophobic amino acids within this NES-like sequence is similar to that of several non-natural high-affinity NESs (*Figure 4B*; [*Engelsma et al., 2004*; *Güttler et al., 2010*]).

To investigate the role of the Mto1 NES-like sequence, we deleted the first 25 amino acids of Mto1 from GFP-Mto1[9A1-NE]-HTB. The truncated protein, termed GFP-Mto1[ΔNES-9A1-NE]-HTB, failed to localize to NPCs and instead was present in the cytoplasm (*Figure 4C*). In parallel, we used LFQ MS to determine how the ΔNES truncation affected the GFP-Mto1[9A1-NE]-HTB interactome. As with LMB treatment, very few proteins were enriched in the GFP-Mto1[9A1-NE]-HTB interactome compared to GFP-Mto1[ΔNES-9A1-NE]-HTB interactome (*Figure 4D*; *Table 3*; *Supplementary file 5*). However, we observed strong enrichment of both Crm1 (~85X) and Nup146 (~20X). The importance of the Mto1 NES-like sequence both for localization to NPCs and for interaction with Crm1 strongly suggests that Mto1 is a direct but unconventional cargo for Crm1. Because of the unusual role of the Mto1 NES-like sequence, we will refer to it as a 'NES-mimic' (NES-M).

## The Mto1 NES-mimic is sufficient for nuclear envelope localization

We next asked whether the Mto1 NES-M is sufficient to localize a reporter protein to the NPC. We replaced endogenous Mto1 with GFP-Mto1[1-29]-GST, which contains only the first 29 amino acids of Mto1. Strikingly, GFP-Mto1[1-29]-GST localized to puncta on the NE, which we interpret to be NPCs (*Figure 4E*). By contrast, GFP-Mto1[1-12]-GST, which lacks the NES-M, did not show any specific localization. We further found that after LMB treatment, GFP-Mto1[1-29]-GST was lost from NPCs (*Figure 4F*); moreover, like Mto1[9A1-NE]-GFP, GFP-Mto1[1-29]-GST was present exclusively in the cytoplasm after LMB treatment.

Compared to GFP fusions with Mto1[NE], GFP-Mto1[1-29]-GST had a weaker punctate localization at NPCs. We hypothesized that this may be due an avidity effect, because Mto1[NE] can form higher order multimers, via its coiled-coil region and via interaction with Mto2 (*Lynch et al., 2014*), whereas GFP-Mto1[1-29]-GST would be expected to form only dimers, via the GST domain. To investigate whether dimerization may contribute to NPC localization, we analyzed localization of a GFP-Mto1[1-29]−13Myc fusion protein, which should be monomeric. GFP-Mto1[1-29]−13Myc did not localize to NPCs (*Figure 4E*), suggesting that dimerization/multimerization may be an important factor for Mto1 NPC localization.

Collectively, these results indicate the Mto1 NES-M is both necessary and sufficient for localization to NPCs, without ever being present in the nucleus.

## Mto1 NPC localization requires RanGTP

To further investigate similarities between the mechanism of Mto1 localization to NPCs and nuclear export, we tested whether Mto1 localization depends on the nucleotide state of Ran. Net directional transport of conventional cargos through the NPC depends on a RanGTP gradient across the NE, generated by Ran GTPase activating protein (RanGAP) in the cytoplasm and Ran guanine-nucleotide exchange factor (RanGEF) in the nucleus (*Aitchison and Rout, 2012*; *Görlich and Kutay, 1999*; *Wente and Rout, 2010*). Importins bind import cargos in the cytoplasm, where RanGTP concentration is low, and release them in the nucleus, where RanGTP concentration is high. By contrast, exportins bind cooperatively to export cargos and RanGTP within the nucleus to form trimeric export complexes, which then dissociate after export, accompanied by RanGTP hydrolysis aided by RanGAP (*Fornerod et al., 1997a*; *Fung and Chook, 2014*; *Güttler and Görlich, 2011*; *Koyama and Matsuura, 2012*; *Monecke et al., 2014*). The role of Ran can be addressed by expressing mutant versions of Ran (encoded by the *spi1+* gene in fission yeast; [*Matsumoto and Beach, 1991*]) that mimic either GTP or GDP states (*Bischoff et al., 1994*; *Klebe et al., 1995*). Constitutively active Ran (RanQ69L in humans) is defective in GTP hydrolysis and thus 'locked' in the RanGTP state, while inactive/dominant-negative Ran (RanT24N in humans) has low affinity for nucleotide and competes with endogenous RanGDP for binding to RanGEF.

We expressed wild-type *spi1+*, *spi1[Q68L]* (equivalent to human RanQ69L), and *spi1[T23N]* (equivalent to human RanT24N) as integrated transgenes from a thiamine-repressible promoter. All

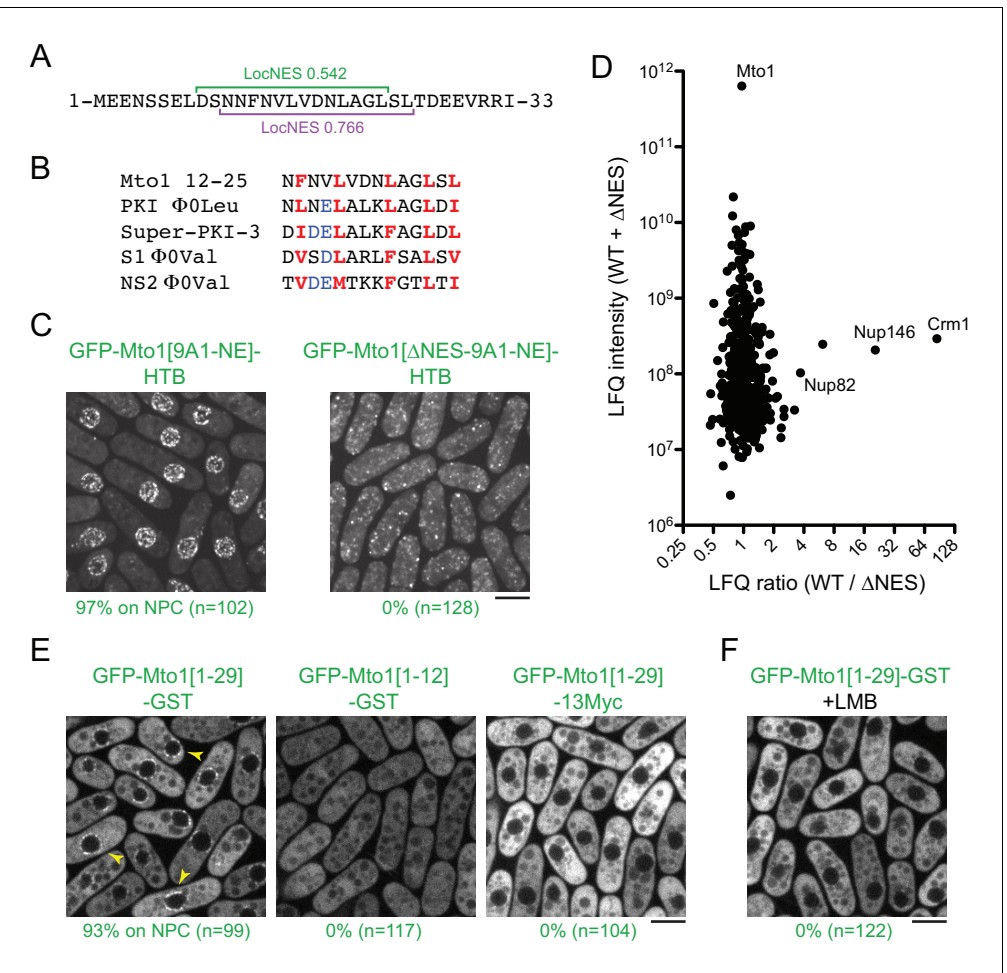

**Figure 4.** Mto1 interacts with Crm1 via a NES-like sequence near the Mto1 amino-terminus. (**A**) Predicted NESs in the first 33 amino acids of Mto1, with associated LocNES scores (*Xu et al., 2015*). These are the only sequences in the first 130 amino acids of Mto1 with LocNES scores greater than 0.1 (**B**) Alignment of Mto1 amino acids 12–25 with four non-natural, high-affinity NESs ('supraphysiological' NESs) described by *Güttler et al. (2010)* and *Engelsma et al. (2004)*. Conserved hydrophic residues are indicated in red. Acidic residues shown to enhance NES affinity for Crm1 are in blue. (**C**) Localization of GFP-Mto1[9A1-NE]-HTB and GFP-Mto1[ΔNES-9A1-NE]-HTB, which lacks Mto1 amino acids 1–25. Numbers below images indicate percent cells with Mto1 at NPCs (n = total number of cells scored). (**D**) Mass spectrometry label-free quantification (LFQ) of 469 proteins from samples of cross-linked, purified GFP-Mto1[9A1-NE]-HTB ('WT') and cross-linked, purified GFP-Mto1[ΔNES-9A1-NE]-HTB ('ΔNES'). 'LFQ ratio' indicates relative enrichment of a given protein in the purified WT sample compared to the purified ΔNES sample. 'LFQ intensity' indicates total intensity (arbitrary units) of a given protein from the combined purified samples. Data shown represent one of two independent biological replicates. Nup82 is labeled because it is likely to interact with Nup146, based on homology with budding yeast (*Belgareh et al., 1998*). See also *Table 3*. Complete datasets are in *Supplementary file 5*. (**E**) Localization of the indicated Mto1 fragments fused to GFP at their N-termini and either GST or 13Myc at their C-termini. Arrowheads indicate examples of localization to the NPCs. Numbers below images indicate percent cells with Mto1 at NPCs (n = total number of cells scored). (**F**) Localization of GFP-Mto1[1-29]-GST in leptomycin B-treated cells. Images in E and F are single Z-sections, while other images are maximum projections. Bars, 5 μm.

DOI: https://doi.org/10.7554/eLife.33465.014

cells were viable under repressing conditions, but growth was impaired by expression of *spi1[Q68L]* or *spi1[T23N]* (*Figure 5—figure supplement 1*), consistent with phenotypes of the equivalent mutants in vertebrate cells (*Clarke et al., 1995*; *Dasso et al., 1994*; *Kornbluth et al., 1994*; *Ren et al., 1994*). To avoid any indirect effects on Mto1[9A1-NE]-GFP localization as a result of growth impairment, we assayed localization as early as possible during expression (*Figure 5*,

**Table 3.** Data for selected proteins from mass spectrometry comparison of GFP-Mto1[9A1-NE]-HTB and GFP-Mto1[ΔNES-9A1-NE]-HTB interactomes.

Peptide counts and label-free quantification (LFQ) values for selected proteins shown in **Figure 4D**. Data from two independent biological replicates are shown. See also **Supplementary file 5**.

| Protein | Replicate 1 (E160419) | | | | | Replicate 2 ((E161127) | | | | |
| | Peptides from strain KS7611 | Peptides from strain KS8573 | LFQ intensity from strain KS7611 | LFQ intensity from strain KS8573 | LFQ ratio | Peptides from strain KS7611 | Peptides from strain KS8573 | LFQ intensity from strain KS7611 | LFQ intensity from strain KS8573 | LFQ ratio |
| --- | --- | --- | --- | --- | --- | --- | --- | --- | --- | --- |
| Alp7 | 15 | 14 | 2.2e8 | 1.4e8 | 1.5 | 12 | 9 | 2.1e8 | 1.4e8 | 1.5 |
| Crm1 | 21 | 2 | 2.9e8 | 3.4e6 | 85.2 | 20 | 2 | 3.5e8 | 5.5e6 | 63.4 |
| Mto1 | 47 | 47 | 3.1e11 | 3.2e11 | 1.0 | 40 | 40 | 3.1e11 | 3.2e11 | 1.0 |
| Mto2 | 10 | 10 | 4.8e9 | 4.1e9 | 1.2 | 8 | 8 | 6.3e9 | 5.9e9 | 1.1 |
| Nsp1 | 10 | 9 | 2.2e8 | 1.2e8 | 1.8 | 10 | 9 | 2.4e8 | 1.0e8 | 2.4 |
| Nup146 | 15 | 3 | 2.0e8 | 9.5e6 | 20.7 | 19 | 0 | 1.5e8 | NQ | NQ |
| Nup82 | 10 | 7 | 8.1e7 | 2.2e7 | 3.7 | 6 | 2 | 5.1e7 | NQ | NQ |
| Plo1 | 17 | 15 | 2.5e8 | 1.6e8 | 1.5 | 17 | 10 | 2.2e8 | 1.8e8 | 1.2 |

NQ = not quantified, because peptide count and/or LFQ intensity in the relevant samples was below threshold for quantification.
DOI: https://doi.org/10.7554/eLife.33465.015

*Figure 5—figure supplement 2*). Expression of *spi1+* had no effect on Mto1[9A1-NE]-GFP localization. Expression of *spi1[Q68L]* impaired import of a nuclear localization signal (NLS) reporter protein, as expected (*Figure 5—figure supplement 2*), but did not alter Mto1[9A1-NE]-GFP localization to NPCs. Interestingly, expression of *spi1[T23N]*, which had only minor effects on NLS reporter localization, led to strong loss of Mto1[9A1-NE]-GFP from NPCs (*Figure 5*, *Figure 5—figure supplement 2*). These results indicate that, like nuclear export, Mto1 localization to NPCs requires RanGTP. Moreover, at least in the short-term, neither RanGDP nor Ran nucleotide cycling is required for Mto1 NPC localization.

## Mto1-Crm1 complex docks at the NPC via Nup146 FG repeats

We next asked whether Nup146 contributes to Mto1 NPC localization. Like approximately one-third of all nucleoporins, Nup146 and its homologs *Sc* Nup159 and *Hs* Nup214 contain multiple phenylalanine-glycine (FG) repeats, which bind directly to importins and/or exportins (*Figure 6—figure supplement 1A*; [*Aitchison and Rout, 2012*; *Wente and Rout, 2010*]). Because of their location on the cytoplasmic face of the NPC, these nucleoporins are classified as 'cytoplasmic FG-Nups', distinguishing them from the 'symmetric FG-Nups' present within the central permeability barrier of the NPC. While FG repeats of symmetric FG-Nups directly facilitate cargo transport through the NPC, FG repeats of cytoplasmic FG-Nups are thought not to be important for transport per se (*Adams et al., 2014*; *Strawn et al., 2004*; *Zeitler and Weis, 2004*), although their other (non-FG) regions recruit proteins for processes linked to transport (e.g. mRNP processing after export (*Napetschnig et al., 2007*; *Schmitt et al., 1999*; *Weirich et al., 2004*); *Figure 6—figure supplement 1A*). Nevertheless, the FG repeats of *Sc* Nup159 and *Hs* Nup214 have been shown to bind to Crm1 with high specificity relative to other importins/exportins. (*Allen et al., 2002*; *Fornerod et al., 1997b*; *Hutten and Kehlenbach, 2006*; *Port et al., 2015*; *Roloff et al., 2013*; *Zeitler and Weis, 2004*). We therefore focused attention on the Nup146 FG repeats.

We deleted the 50-amino-acid region comprising FG repeats 5–12 (out of a total of 16 FG repeats) from the endogenous *nup146* coding sequence (*Figure 6A*). Although complete deletion of *nup146* is lethal (*Chen et al., 2004*), the *nup146[ΔFG5-12]* strain was viable, and Nup146[ΔFG5-12]—3mCherry was localized to NPCs. (*Figure 6—figure supplement 1B*). Strikingly, in *nup146[ΔFG5-12]* cells, Mto1[9A1-NE]-GFP no longer localized to NPCs and instead was present only in the cytoplasm (*Figure 6B*; *Figure 6—figure supplement 1B*).

We also analyzed MTOC activity at the NE in wild-type (*nup146+*) cells vs. *nup146 [ΔFG5-12]* cells. First, we assayed MT regrowth after cold-induced depolymerization, in cells expressing full-length, untagged Mto1 (*Figure 6C*). In wild-type cells, MT regrowth occurred from the NE, while in

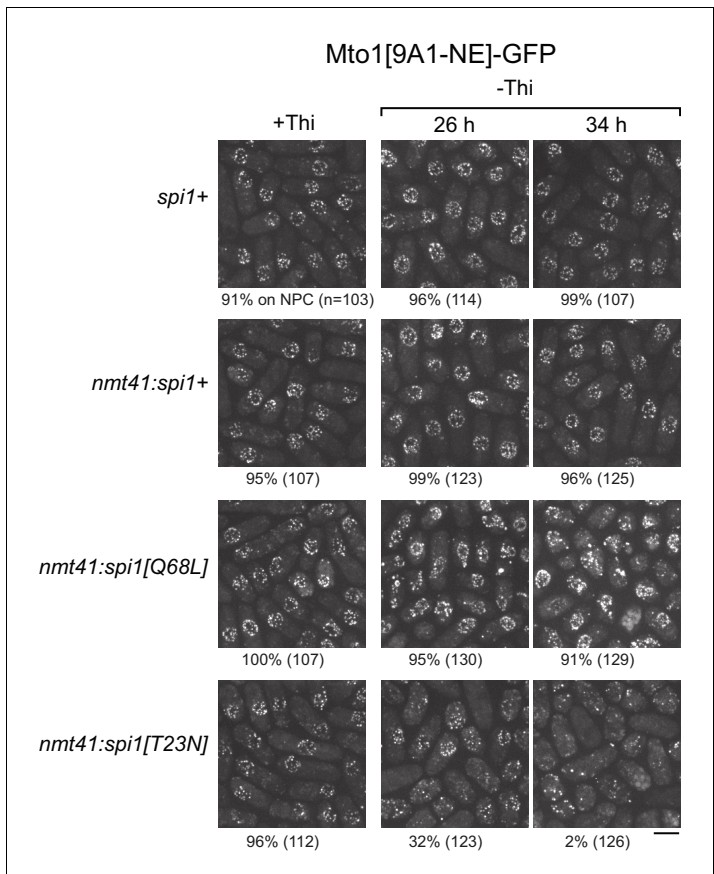

**Figure 5.** Expression of dominant-negative Ran (spi1[T23N]) but not constitutively active Ran (spi1[Q68L]) disrupts localization of Mto1[NE] to nuclear pore complexes. Mto1[9A1-NE]-GFP localization in strains containing different versions of Ran (*spi1* in fission yeast) expressed from the thiamine-repressible *nmt41* promoter, together with control wild-type cells (*spi1+*). Cells are shown in the presence of thiamine (+Thi), and 26 and 34 hr after removal of thiamine (-Thi). 26 and 34 hr represent early and later stages of induced expression, respectively (see *Figure 5—figure supplement 2*). Numbers below images indicate percent cells with Mto1[9A1-NE]-GFP on nuclear pore complexes (n = total number of cells scored). Bar, 5 µm.

DOI: https://doi.org/10.7554/eLife.33465.016

The following figure supplements are available for figure 5:

**Figure supplement 1.** Effects of mutant Ran (*spi1* in fission yeast) on cell viability.
DOI: https://doi.org/10.7554/eLife.33465.017

**Figure supplement 2.** Effects of mutant Ran (*spi1* in fission yeast) on Mto1[NE] localization to nuclear pore complexes and on import of a nuclear localization signal (NLS) reporter.
DOI: https://doi.org/10.7554/eLife.33465.018

*nup146[ΔFG5-12]* cells MT regrowth occurred randomly in the cytoplasm (*Figure 6C*), similar to LMB-treated wild-type cells (*Figure 3—figure supplement 3*). Second, we used live-cell imaging of GFP-tubulin to assay steady-state MT nucleation, in cells expressing Mto1[NE]-GFP (*Figure 6D,E*; in these cells, Mto1[NE]-GFP is too faint to be seen relative to GFP-tubulin). In *nup146[ΔFG 5–12]* cells, MT nucleation frequency in the vicinity of the NE was decreased by ~90% relative to wild-type (*nup146+*) cells, while nucleation frequency away from the NE was unchanged.

Collectively, these results indicate that Nup146 FG repeats 5–12 are essential for Mto1 docking at the NPC and, consequently, for MTOC nucleation from the NE. To our knowledge, this is the first biological function that can be uniquely attributed to the FG repeats of the Nup146/*Sc* Nup159/*Hs* Nup214 class of cytoplasmic FG-Nups, in any organism (see Discussion).

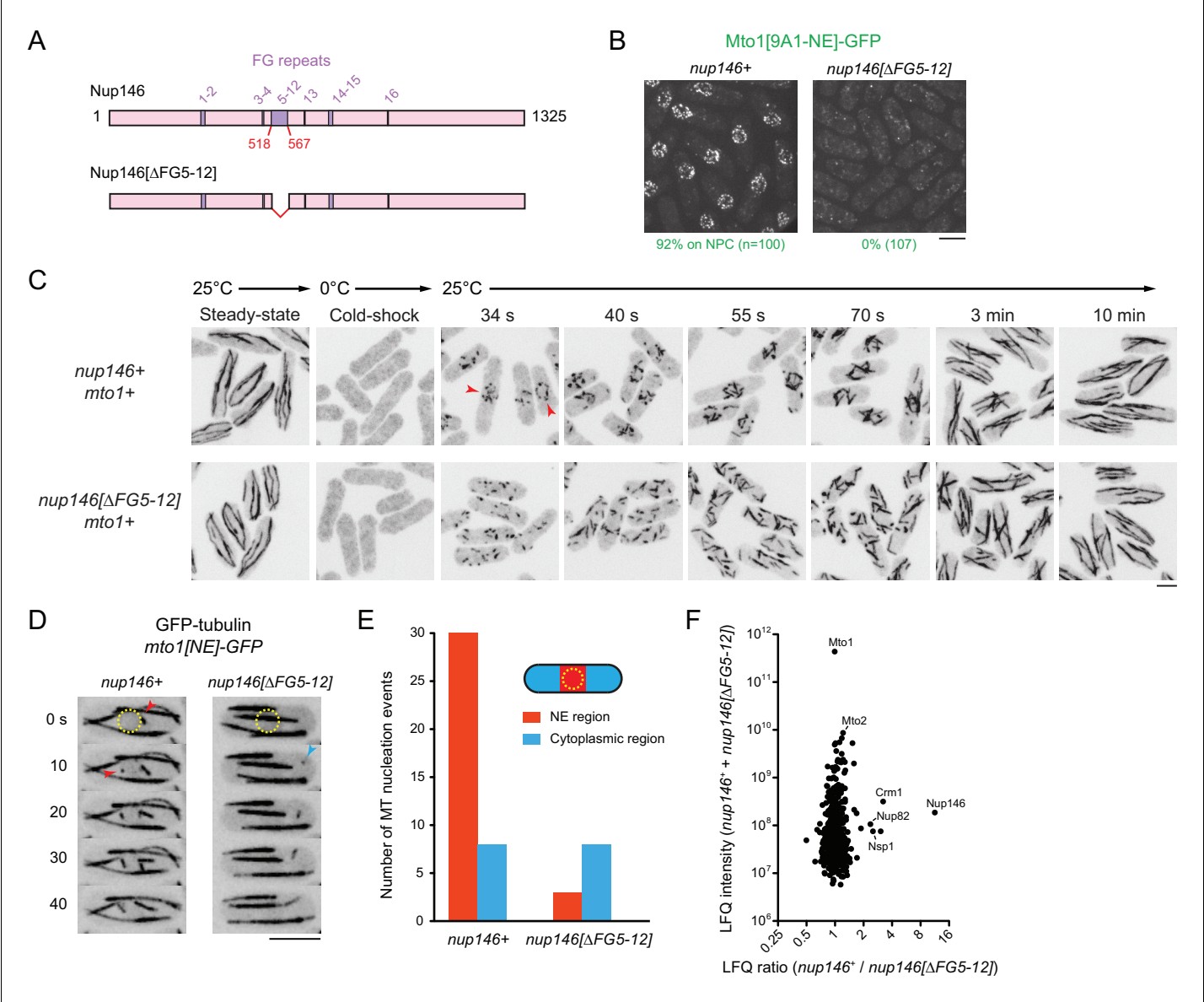

**Figure 6.** Nup146 FG repeats are required for Mto1[NE] docking at nuclear pore complexes, microtubule nucleation from the nuclear envelope region, and Mto1 interaction with Nup146. (**A**) Diagram of Nup146 and Nup146[ΔFG5-12]. (**B**) Localization of Mto1[9A1-NE]-GFP in wild-type (*nup146+*) and *nup146[ΔFG5-12]* cells. Numbers below images indicate percent cells with Mto1 on NPCs (n = total number of cells scored). (**C**) Anti-tubulin immunofluorescence of wild-type (*nup146+*) and *nup146[ΔFG5-12]* cells during microtubule (MT) regrowth after cold-induced MT depolymerization. Both strains express full-length, wild-type Mto1. Arrowheads in *nup146+* cells indicate examples of MT regrowth from the nuclear envelope (NE), which does not occur in *nup146[ΔFG5-12]* cells. (**D**) GFP-tubulin images from time-lapse video showing MT nucleation in wild-type (*nup146+*) and *nup146 [ΔFG5-12]* cells. Yellow dashed line indicates cell nucleus. Red arrowheads indicate nucleation from the NE region. Blue arrowhead indicates nucleation from non-NE cytoplasmic region. In these cells, Mto1[NE] is also tagged with GFP but is too faint to be seen relative to GFP-tubulin. (**E**) Quantification of MT nucleation from videos of the type shown in D. Numbers represent total number of events for 90 cells of each strain, imaged for 100 s. Differences between strains were highly significant (p=0.0026; Fisher's exact test, two-sided). (**F**) Mass spectrometry label-free quantification (LFQ) of 512 proteins from samples of cross-linked, purified GFP-Mto1[9A1-NE]-HTB from wild-type (*nup146+*) and from *nup146[ΔFG5-12]* cells. 'LFQ ratio' indicates relative enrichment of a given protein in the purified sample from *nup146+* cells compared to the purified sample from *nup146[ΔFG5-12]* cells. 'LFQ intensity' indicates total intensity (arbitrary units) of a given protein from the combined purified samples. Data shown represent geometric mean from two independent biological replicates. Nup82 and Nsp1 are labeled because they are likely to interact with Nup146, based on homology with budding yeast (***Belgareh et al., 1998***). See also *Table 4*. Complete datasets are in ***Supplementary file 6***. Bars, 5 μm.

DOI: https://doi.org/10.7554/eLife.33465.019

The following figure supplement is available for figure 6:

**Figure supplement 1.** Additional characterization of Nup146[ΔFG5-12].

*Figure 6 continued on next page*

*Figure 6 continued*

DOI: https://doi.org/10.7554/eLife.33465.020

## Nup146 FG repeats stabilize the Mto1-Crm1 interaction

Our results thus far indicate that a RanGTP-dependent Mto1-Crm1 'cargo-like' complex docks at the cytoplasmic face of the NPC via a mechanism involving Nup146 FG repeats (see *Figure 7*). Interestingly, a subset of FG repeats in *Hs* Nup214 have been shown to bind to Crm1 in a manner that stabilizes the Crm1-RanGTP-cargo interaction in vitro (*Askjaer et al., 1999*; *Fornerod et al., 1997b*; *Hutten and Kehlenbach, 2006*; *Kehlenbach et al., 1999*; *Port et al., 2015*; *Roloff et al., 2013*). We therefore asked whether Nup146 FG repeats 5–12 are important for Mto1 interaction with Nup146, and whether these repeats contribute to Crm1 association with Mto1 in vivo. We used LFQ MS to compare GFP-Mto1[9A1-NE]-HTB interactomes from wild-type (*nup146+*) vs. *nup146[ΔFG5-12]* cells. Among more than 500 quantified proteins, only five to six proteins were significantly enriched in the GFP-Mto1[9A1-NE]-HTB interactome from *nup146+* cells compared to *nup146*

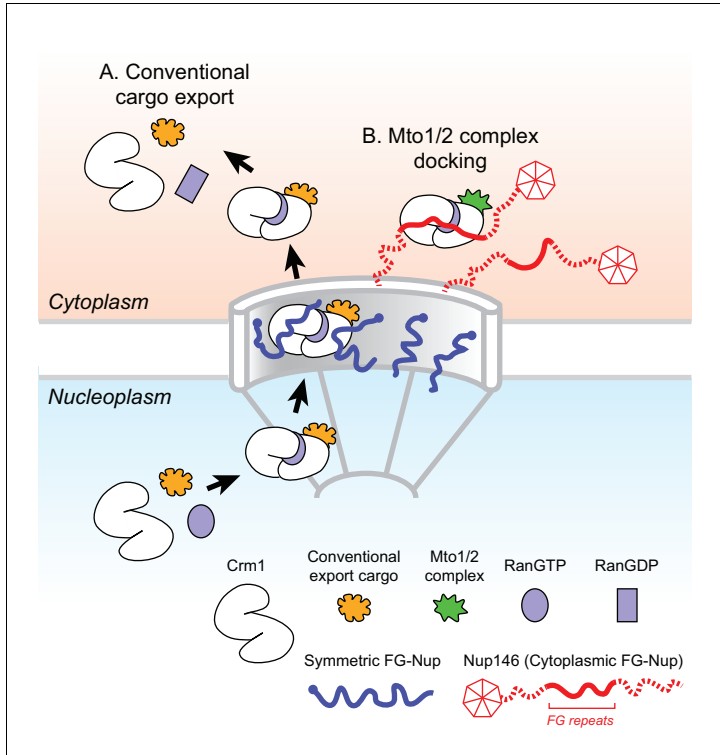

**Figure 7.** Model for Mto1/2 complex docking at the nuclear pore complex (NPC), compared to conventional nuclear export. (**A**) Conventional export cargos form a trimeric complex with Crm1 and RanGTP in the nucleus. Passage through NPC permeability barrier depends on interaction of export complexes with FG repeats of symmetric FG-Nups. While passing through the NPC, some export complexes may also interact with FG repeats of Nup146 (not shown). Once in the cytoplasm, export complexes are disassembled by soluble RanBP1 and RanGAP, and RanGTP is hydrolyzed to RanGDP (here, multiple steps are simplified to a single step). (**B**) Mto1/2 complex docked at NPC. Mto1/2 is sourced from a cytoplasmic pool rather than a nucleoplasmic pool. The Mto1 NES-M binds Crm1 by mimicking an export cargo, and the docking complex binds to cytoplasmic FG-Nup Nup146. Docking requires RanGTP and FG repeats of Nup146. See also *Figure 7—figure supplement 1*.
DOI: https://doi.org/10.7554/eLife.33465.021

The following figure supplement is available for figure 7:

**Figure supplement 1.** Models for stable docking of a high-affinity NES cargo at the cytoplasmic face of the nuclear pore complex via Nup146 and for formation of export-like complexes from cytoplasmic cargo.
DOI: https://doi.org/10.7554/eLife.33465.022

[ΔFG5-12] cells. Nup146 itself showed the greatest enrichment (~11X), while Crm1 was also enriched, although to a lesser extent (~3X) (*Figure 6F*; *Table 4*; *Supplementary file 6*). This suggests that Nup146 FG repeats are essential for interaction of Mto1 with Nup146. In addition, while Nup146 FG repeats may not be absolutely essential for formation of an Mto1-Crm1 complex, they may help to stabilize it.

## Discussion

While different mechanisms are involved in generating non-centrosomal MTOCs at different subcellular sites, at many sites the mechanisms themselves remain poorly understood (*Petry and Vale, 2015*; *Sanchez and Feldman, 2017*; *Wu and Akhmanova, 2017*). Here, we have shown how MTOCs are generated at the NE in fission yeast *S. pombe* via the Mto1/2 complex. We find that Mto1 docks at the cytoplasmic face of NPCs, and this involves a novel mechanism in which the nuclear export machinery at NPCs is repurposed for a non-export-related function. Docking depends on an export cargo-like interaction between a NES-like sequence (NES-M) at the Mto1 N-terminus and the NES-binding cleft of exportin Crm1, the major transport receptor for protein nuclear export (*Fung and Chook, 2014*; *Hutten and Kehlenbach, 2007*; *Kutay and Güttinger, 2005*). Docking further requires RanGTP and the central FG repeats of Nup146, a cytoplasmic FG-Nup homologous to *Sc* Nup159 and *Hs* Nup214. The general features of Mto1 docking at NPCs are summarized in *Figure 7*.

In this work, chemical cross-linking of cell cryogrindates allowed us to capture low-affinity interactions that might otherwise be unstable during conventional purification. By using affinity tags compatible with strong denaturing conditions (*Tagwerker et al., 2006*), we were able to solubilize and interrogate protein-protein interactions that normally occur within 'solid-phase' subcellular environments. In addition, by combining live-cell microscopy with LFQ MS (*Cox and Mann, 2008*) we were able to correlate changes in Mto1 localization with changes in interactors on a near proteome-wide scale, and under several different comparative conditions.

The mechanism described here for Mto1 localization to NPCs was entirely unexpected. While it incorporates many of the elements of conventional Crm1-dependent nuclear export (*Figure 7*), there are two fundamental distinctions. First, when not interacting with the export machinery, Mto1(and its partner Mto2) is a cytoplasmic protein rather than a nuclear protein. Second, interaction of Mto1 with the export machinery leads to docking at NPCs, rather than a return/release into the cytoplasm. To our knowledge, this is the first example of a nuclear export-like complex being used to dock a cytoplasmic 'cargo' at the NPC, with no obvious functional link to export.

**Table 4.** Data for selected proteins from mass spectrometry comparison of GFP-Mto1[9A1-NE]-HTB interactomes in wild-type (*nup146+*) and *nup146[ΔFG5-12]* cells.

Peptide counts and label-free quantification (LFQ) values for selected proteins from the two replicate experiments contributing to the graph in *Figure 6F*. Complete datasets are in *Supplementary file 6*.

| Protein | Replicate 1 (E170214) | | | | | Replicate 2 (E170306) | | | | |
|---|---|---|---|---|---|---|---|---|---|---|
| | Peptides from strain KS9021 | Peptides from strain KS9077 | LFQ intensity from strain KS9021 | LFQ intensity from strain KS9077 | LFQ ratio | Peptides from strain KS9021 | Peptides from strain KS9077 | LFQ intensity from strain KS9021 | LFQ intensity from strain KS9077 | LFQ ratio |
| Alp7 | 17 | 15 | 1.4e8 | 1.4e8 | 1.0 | 16 | 16 | 1.0e8 | 8.2e7 | 1.3 |
| Crm1 | 31 | 17 | 3.4e8 | 9.2e7 | 3.6 | 27 | 18 | 1.7e8 | 6.1e7 | 2.9 |
| Mto1 | 62 | 61 | 2.7e11 | 2.6e11 | 1.0 | 55 | 54 | 1.7e11 | 1.8e11 | 0.9 |
| Mto2 | 12 | 12 | 6.0e9 | 5.0e9 | 1.2 | 13 | 13 | 3.8e9 | 3.1e9 | 1.2 |
| Nsp1 | 11 | 3 | 8.3e7 | 2.7e7 | 3.0 | 9 | 4 | 3.5e7 | 1.7e7 | 2.1 |
| Nup146 | 29 | 6 | 2.1e8 | 1.5e7 | 13.9 | 31 | 10 | 1.4e8 | 1.5e7 | 9.2 |
| Nup82 | 13 | 10 | 1.1e8 | 4.0e7 | 2.8 | 12 | 11 | 4.9e7 | 2.5e7 | 2.0 |
| Plo1 | 17 | 17 | 1.5e8 | 1.3e8 | 1.1 | 17 | 17 | 1.1e8 | 8.1e7 | 1.3 |

DOI: https://doi.org/10.7554/eLife.33465.023

Nuclear transport receptors and nucleoporins are known to have non-transport-related roles away from NPCs, especially in relation to mitosis and microtubule function. In many cell types, importins play a major role in RanGTP-dependent regulation of mitotic spindle assembly factors, while Crm1 has been described to have roles both at centrosomes and kinetochores (reviewed in (*Cavazza and Vernos, 2015*; *Forbes et al., 2015*; *Kalab and Heald, 2008*)). In particular, in mammalian cells, Crm1 is involved in RanGTP-dependent kinetochore targeting of a complex containing RanGAP1 and RanBP2 (a metazoan-specific cytoplasmic FG-Nup, also known as Nup358,), and this is important for the integrity of kinetochore microtubules (*Arnaoutov et al., 2005*), perhaps because of the SUMO E3 ligase activity of RanBP2 (*Ritterhoff et al., 2016*). In this context, it is interesting to compare and contrast the role of Crm1 at the kinetochore with our proposed mechanism of Mto1 docking. While the role of Crm1 in Mto1 docking is to recruit a cargo-like molecule (i.e. Mto1) to NPCs, the role of Crm1 at the kinetochore is to recruit specific NPC components to a novel non-NPC site (likely via RanBP2's FG repeats; [*Ritterhoff et al., 2016*]). In Crm1-dependent targeting of RanGAP1/RanBP2 to kinetochores, the Crm1 'cargo' remains unknown, and the mechanism by which Crm1 associates with kinetochores appears to be controversial (*Platani et al., 2009*; *Zuccolo et al., 2007*). We further note that recruitment of nucleoporins to the kinetochore is not restricted to the RanGAP1/RanBP2 complex. In many cell types, the multi-protein Nup107-160 complex (also known as Y complex; [*Kabachinski and Schwartz, 2015*]) is also recruited to kinetochores, where it may promote MT nucleation by the γ-tubulin complex (*Mishra et al., 2010*). This process also depends on RanGTP, but any specific role for nuclear transport receptors here remains unclear.

## Docking at the NPC

How does Mto1, a nuclear export cargo 'mimic', become docked at the NPC, while conventional export cargos are released into the cytoplasm? Ultimately, a detailed understanding of this issue will require in vitro biochemistry with purified proteins. However, based on previous work in mammalian cells (*Engelsma et al., 2004*; *Port et al., 2015*), we speculate that docking may depend on: 1) the Mto1 NES-M acting as a high-affinity NES; and 2) the stability of interaction between Mto1-Crm1 and Nup146 FG repeats.

The Mto1 NES-M is necessary and sufficient for docking at the NPC (*Figure 4*). Interestingly, in human cells, cargo containing a non-natural, high-affinity NES (a 'supraphysiological NES') was shown to accumulate at the cytoplasmic face of the NPC and also to enhance Crm1 accumulation at the same site (*Engelsma et al., 2004*; *Engelsma et al., 2008*). We hypothesize that the Mto1 NES-M may be a *natural* high-affinity NES. In recent years, the NES 'consensus' has evolved in concert with new experimental findings (*Dong et al., 2009*; *Fung et al., 2015*; *Fung et al., 2017*; *Güttler et al., 2010*; *Monecke et al., 2009*). In particular, relative to an original consensus involving four spaced hydrophobic residues (*Kutay and Güttinger, 2005*), several high-affinity NESs depend on a fifth hydrophobic residue, which may also be present in the Mto1 NES-M (*Figure 4B*). In this context, it is interesting that we found that Mto1[9A1-NE]-GFP localizes to NPCs in *crm1-C529A* mutants (*Figure 3B*) but not in *crm1-C529S, crm1-C529T, and crm1-C529V* mutants, even though these mutants are viable and thus competent for nuclear export (*Figure 3—figure supplement 2*). This may indicate that, relative to conventional NESs, the binding of the Mto1 NES-M to Crm1 involves recognition of additional and/or distinct features within the Crm1 NES-binding cleft.

Assuming that the Mto1 NES-M interacts with Crm1 as a high-affinity NES, clues as to how this could lead to accumulation at the NPC can be found in structural studies of Crm1 alone and Crm1 in complex with RanGTP, cargo, and an FG-repeat fragment of *Hs* Nup214 (*Figure 7—figure supplement 1A*; [*Dong et al., 2009*; *Güttler et al., 2010*; *Monecke et al., 2009*; *Monecke et al., 2013*; *Port et al., 2015*; *Saito and Matsuura, 2013*]). Crm1 can exist in two conformations: an unliganded extended, superhelical conformation, which is inhibitory to cargo and RanGTP binding, and a compact, ring-like conformation, which is stabilized by cooperative binding to cargo and RanGTP. Importantly, the FG-repeat fragment of *Hs* Nup214, which binds Crm1 cooperatively with RanGTP and cargo, interacts with the compact conformation of Crm1 at multiple sites, spanning the junction between the Crm1 N- and C-termini in a manner similar to an adhesive bandage (*Port et al., 2015*) (*Figure 7—figure supplement 1A*). The *Hs* Nup214 FG repeats have therefore been described as a 'molecular clamp' that can stabilize Crm1-RanGTP-cargo complex in the compact conformation (*Port et al., 2015*). However, from an energetic perspective, cooperative binding also implies that anything that stabilizes the Crm1 compact conformation (including a high-affinity NES) will

correspondingly reinforce association of Crm1 with *Hs* Nup214 FG repeats. As a result, a sufficiently high-affinity NES cargo would be expected to stabilize interaction of Crm1 with Nup146, leading to docking of Crm1 (and the NES cargo itself) at the cytoplasmic face of the NPC (*Figure 7—figure supplement 1A*).

In addition to a 'high-affinity NES' mechanism, other factors may also contribute to docking of the Mto1/2 complex at the NPC. For example, if Mto1(or its partner Mto2) were to bind Nup146 independently of binding to Crm1, such multivalent binding would decrease the off-rate from the NPC; currently our MS data cannot distinguish between direct and indirect Mto1 interactors. Interactions between Mto1/2 and the NPC could also be stabilized by avidity effects (*Figure 4E*). Mto1/2 is multimeric in vivo, containing multiple (>10) copies of both Mto1 and Mto2 (*Lynch et al., 2014*), while nucleoporins are also present in multiple copies within the NPC, because of its eight-fold symmetry (*Aitchison and Rout, 2012*; *Görlich and Kutay, 1999*; *Wente and Rout, 2010*). As a result, multiple Mto1 molecules in a single Mto1/2 complex could bind to multiple nucleoporins (and/or Crm1) in a single NPC. Interestingly, localization of Mto1/2 to the SPB and the CAR also depends on avidity effects (*Samejima et al., 2010*).

## Formation of an Crm1-dependent docking complex with a cytoplasmic 'cargo'

Given that conventional Crm1-dependent export complexes form in the nucleus, where RanGTP concentration is high (*Aitchison and Rout, 2012*; *Görlich and Kutay, 1999*; *Wente and Rout, 2010*), how might an Mto1/2 docking complex form in the cytoplasm, where RanGTP concentration is low? We speculate that if the Mto1 NES-M acts as a high-affinity NES, it may be possible for Mto1/2 to replace a conventional nuclear export cargo at the final stages of export, via a 'cargo-handover' mechanism (*Figure 7—figure supplement 1B*). Alternatively, a docking complex involving Mto1/2, Crm1, Nup146 and RanGTP could in principle form de novo at the cytoplasmic face of the NPC. While the low concentration of RanGTP in the cytoplasm makes this unlikely, it is formally possible that in the immediate vicinity of the NPC, the local concentration of RanGTP is higher than in the cytoplasm in general, because in yeast, RanGAP is freely soluble in the cytoplasm rather than associated with the NPC (*Aitchison and Rout, 2012*; *Hopper et al., 1990*). Accordingly, immediately after RanGTP dissociates from export complexes (but prior to GTP hydrolysis), it might be available to cytoplasmic Mto1/2.

Our proposition that an Mto1/2 docking complex includes RanGTP (see *Figure 7*; *Figure 7—figure supplement 1*) is based on the observed requirement for RanGTP for docking in vivo and on analogy to the known mechanisms underlying stable NES-dependent binding of conventional nuclear export cargo to Crm1 (*Dong et al., 2009*; *Güttler et al., 2010*; *Monecke et al., 2009*; *Monecke et al., 2013*; *Port et al., 2015*; *Saito and Matsuura, 2013*). We note that in cross-linking MS experiments, we did not observe increased association of Ran with Mto1 that is localized to the NE compared to Mto1 that is not localized to the NE (*Supplementary file 3–6*. One possible reason for this is that Ran may not be readily cross-linked to Mto1 or to Mto1's immediate interactors (by analogy to conventional cargo export, Ran would not be expected to bind directly to Mto1). Alternatively, it is possible that the Mto1/2 docking complex does not contain Ran and that the requirement for Ran-GTP for docking is only indirect. This might be the case if the main role of Ran-GTP in Mto1/2 docking is to generate sufficiently high levels of conventional export complexes at the cytoplasmic face NPC such that components of these complexes (e.g. Crm1) can subsequently be used for Mto1/2 docking by an unconventional, Ran-independent mechanism. Further work with purified proteins may help to address this issue.

## A novel function for cytoplasmic FG-Nups?

In this work, we identified a very specific phenotype associated with deletion of Nup146 FG repeats 5–12: Mto1 is lost from NPCs, with a concomitant loss in MT nucleation from the NE. Moreover, this is correlated with a strong decrease in interaction of Mto1 with Nup146 and, to a lesser extent, with Crm1, consistent with our model of a cargo-like complex of Mto1/2, Crm1 and RanGTP docking at the cytoplasmic face of the NPC. In this context, it is interesting that extensive analysis in budding yeast has shown that the FG regions of cytoplasmic FG-Nups (as well as nucleoplasmic FG-Nups) can be deleted without almost any discernible effects on nuclear transport (*Adams et al., 2014*;

*Strawn et al., 2004*; *Zeitler and Weis, 2004*). In human cells, the role of *Hs* Nup214 in protein export appears to be somewhat controversial (*Bernad et al., 2006*; *Hutten and Kehlenbach, 2006*); however, similar to budding yeast, in at least one instance where *Hs* Nup214 was found to be important for export—namely, export of the 60S pre-ribosome— the FG repeats of *Hs* Nup214 were found not to be required [*Bernad et al., 2006*]). Based on these results, and on the conservation of FG repeats in Nup146, *Sc* Nup159 and *Hs* Nup214, we propose that an important but previously unrecognized role for cytoplasmic FG-Nups may be to dock cytoplasmic proteins at the NPC for non-export-related functions, as described here for generation of non-centrosomal MTOCs by the Mto1/2 complex. It will be interesting to see how widespread this type of repurposing of the nuclear export machinery is in eukaryotic cells more generally.

# Materials and methods

**Key resources table**

| Reagent type (species) or resource | Designation | Source or reference | Identifiers | Additional information |
|---|---|---|---|---|
| Gene (*Schizosaccharomyces pombe*) | Mto1 | NA | Pombase:SPCC417.07c | |
| Gene (*S. pombe*) | Mto2 | NA | Pombase:SPBC902.06 | |
| Gene (*S. pombe*) | Crm1 | NA | Pombase:SPAC1805.17 | |
| Gene (*S. pombe*) | Nup146 | NA | Pombase:SPAC23D3.06c | |
| Gene (*S. pombe*) | Alp7 | NA | Pombase:SPAC890.02c | |
| Gene (*S. pombe*) | Nup132 | NA | Pombase:SPAC1805.04 | |
| Gene (*S. pombe*) | Spi1 | NA | Pombase:SPBC1289.03c | Spi1 is the *S. pombe* gene name for Ran GTPase |
| Gene (*S. pombe*) | Plo1 | NA | Pombase:SPAC23C11.16 | |
| Gene (*S. pombe*) | Alp14 | NA | Pombase:SPCC895.07 | |
| Gene (*Saccharomyces cerevisiae*) | Nup159 | NA | SGD:S000001377 | Alternate systematic name YIL115C |
| Gene (*Homo sapiens*) | Nup214 | NA | HGNC:8064 | Alternate name CAN |
| Antibody | TAT1 anti-tubulin (mouse monoclonal) | PMID:2606940 | | (1:15 of hybridoma supernatant) |
| Antibody | Alexa Fluor 488 secondary (donkey anti-mouse polyclonal) | Thermo Fisher Scientific | Thermo Fisher Scientific :A-21202; RRID:AB_141607 | (1:80) |
| Antibody | anti-GFP (rabbit polyclonal) | Rockland | Rockland:600-401-215; RRID:AB_828167 | (1:400) |
| Antibody | Alexa 594 FluoroNanogold Fab' fragment (goat anti-rabbit polyclonal) | Nanoprobes | Nanoprobes:7304 | (1:400) |
| Antibody | anti-GFP (sheep polyclonal) | other | | homemade lab stock; used for immunoprecipitation |
| Commercial assay or kit | QuikChange II Site Directed Mutagenesis Kit | Agilent | Agilent:200523 | |
| Chemical compound, drug | Leptomycin B | LC Laboratories | LC Laboratories: L-6100 | |
| Software, algorithm | Metamorph | Molecular Devices | RRID:SCR_002368 | |
| Software, algorithm | Image J | NIH | RRID:SCR_003070 | |
| Software, algorithm | Prism | Graphpad | RRID:SCR_002798 | |

*Continued on next page*

*Continued*

| Reagent type (species) or resource | Designation | Source or reference | Identifiers | Additional information |
|---|---|---|---|---|
| Software, algorithm | Photoshop | Adobe | RRID:SCR_014199 | |
| Software, algorithm | Illustrator | Adobe | RRID:SCR_010279 | |
| Software, algorithm | MaxQuant | PMID:19029910 | RRID:SCR_014485 | |
| Software, algorithm | MaxLFQ | PMID:24942700 | | |
| Software, algorithm | Pombase | PMID:25361970 | RRID:SCR_006586 | |
| Other | DAPI | Thermo Fisher Scientific | Thermo Fisher Scientific: D1306; RRID:AB_2629482 | |
| Other | Protein G Dynabeads | Thermo Fisher Scientific | Thermo Fisher Scientific: 10003D | |
| Other | Fractogel EMD Chelate (M) | Merck KGaA | Merck:1.10338.0010 | |
| Other | Nanolink streptavidin magnetic beads | Trilink | Trilink:M-1002 | |

## Yeast cultures, strain and plasmid construction

Fission yeast methods and growth media were as described (*Forsburg and Rhind, 2006*; *Petersen and Russell, 2016*). Strains were normally grown in YE5S-rich medium or PMG minimal medium (like EMM2, but using 5 g/L sodium glutamate acid instead of ammonium chloride as nitrogen source). For preliminary experiments using SILAC mass spectrometry, cells were grown in low-nitrogen EMM2 medium ('LowN'; using 0.3 g/L ammonium chloride as nitrogen source; [*Bicho et al., 2010*]). For purification of HTB-tagged Mto1 variants for LFQ MS, Mto1 variants were expressed from the *nmt81* promoter, and cells were grown in PMG medium, except for experiments involving leptomycin B (*Figure 3*), in which case Mto1 variants were expressed from the repressed *nmt1* promoter, and cells were grown in 4xYE5S medium. For electron microscopy, cells were grown in EMM2 minimal medium. Nutritional supplements were normally used at 175 mg/L, except for arginine and lysine in SILAC experiments, in which unlabeled arginine or L-$^{13}C_6$-arginine (Sigma Isotec, Gillingham, UK) was used at 80 mg/L, and unlabeled lysine or L-$^{13}C_6$$^{15}N_2$-lysine (Sigma Isotec) was used at 60 mg/L (*Bicho et al., 2010*). Solid media contained 2% Bacto agar (Becton Dickinson, Wokingham, UK). For mating, SPA plates containing 45 mg/L each of adenine, leucine, uracil, histidine and lysine were used. For repression of thiamine-regulated promoters, sterile-filtered thiamine was added to media at a final concentration of 15 μM.

Strains used in this study are listed in *Supplementary file 1*. For experiments purifying HTB-tagged Mto1 for LFQ MS, strains contained the *mto2[17A]* allele; this allele contains 17 phosphorylation sites in Mto2 mutated to alanine, which helps to stabilize the Mto1/2 complex (*Borek et al., 2015*). The *mto2[17A]* allele was also present in strains imaged in *Figures 1C* and *4C* (see *Supplementary file 1*).

Genetic crosses used either tetrad dissection or random spore analysis (*Ekwall and Thon, 2017*). Except for the cases described below, genome manipulations such as gene tagging, truncation and/or deletion were made by homologous recombination of PCR products (*Bähler et al., 1998*; *Hentges et al., 2005*; *Van Driessche et al., 2005*). PCR was performed with either Phusion High-Fidelity polymerase or Q5 High-Fidelity polymerase (NEB, Hitchin, UK). Desired strains were confirmed by yeast colony PCR, western blot and/or fluorescence microscopy as appropriate. For all cloning experiments, *E. coli* strain DH5alpha was used.

## Leptomycin-resistant crm1 mutants

To generate *crm1-C529A/S/T/V* mutants, a one-step approach was used, in which *mto1[9A1-NE]-GFP nup146-3mCherry* cells were transformed with mutated *crm1* DNA fragments and selected directly for leptomycin (LMB) resistance. Mutant *crm1* fragments were designed with the mutation

site in the center, ~650 base pairs of *crm1* sequence on either side of the mutation site, and BstXI sites at each end of fragment. Plasmids containing the mutant fragments were synthesized by GeneArt (plasmids pKS1735, pKS1734, pKS1737, and pKS1738, for *crm1-C529A, crm1-C529S, crm1-C529T,* and *crm1-C529V* mutants, respectively). The *crm1* fragments were released from plasmids by BstXI digestion, purified and transformed into strain KS7255. Cells from the transformation were plated onto YE5S plates containing 300 nM LMB (LC Laboratories, Woburn, MA), and LMB-resistant colonies were easily identified. A negative-control transformation conducted in parallel did not yield any LMB-resistant colonies. Stable LMB-resistant colonies from each transformation were then used for sequencing to confirm specific mutations in crm1 genomic DNA. The mutant strains were named KS9340 (*crm1-C529A*), KS9221 (*crm1-C529S*) KS9338 (*crm1-C529T*), and KS9336 (*crm1-C529V*).

## Overexpression of wild-type and mutant spi1

Strains overexpressing *spi1+, spi1[Q68L]* and *spi1[T23N]* from the *nmt41* promoter were generated by targeted integration of transgenes at the *hph.171k* locus (*Fennessy et al., 2014*). First, pJET-spi1 +/[Q68L]/[T23N] plasmids were constructed. To construct pJET-*spi1+*, *spi1+* genomic DNA was amplified from fission yeast genomic DNA using primer pair OKS3290/OKS3291, and the PCR product was ligated into vector pJET1.2 (Thermo Fisher Scientific, Paisley, UK). The resulting pJET-*spi1+* plasmid was confirmed by sequencing and named pKS1603. To construct pJET-spi1*[Q68L],* the *Q68L* mutation was introduced into pKS1603 by PCR, using primer pair OKS3139/OKS3140. The PCR product was recircularized using T4 polynucleotide kinase and T4 DNA ligase. The resulting plasmid was confirmed by sequencing and named pKS1596. To construct pJET-*spi1[T23N]*, pKS1603 was used as template to introduce the *T23N* mutation into the *spi1* sequence, using QuikChange II Site-Directed Mutagenesis kit (Agilent, Stockport, UK) and primer pair OKS 3336/OKS3337. After DpnI treatment and transformation, the resulting plasmid was confirmed by sequencing and named pKS1595.

Next, the *spi1* inserts from pKS1603, pKS1596, and pKS1595 were each subcloned into the fission yeast integration vector pINTH41 (*Fennessy et al., 2014*) after restriction digest with BamHI and NdeI. The resulting pINTH41-*spi1+/[Q68L]/[T23N]* plasmids were confirmed by restriction digest and named pKS1597, pKS1599, and pKS1598, respectively.

For transformation into fission yeast, pKS1597, pKS1599, and pKS1598 were digested with NotI, and the relevant fragments were purified and used to transform strain KS 7742. Stable nourseothricin-resistant, hygromycin-sensitive integrants were identified, indicating replacement of the hygromycin-resistance marker at the *hph.171k* locus by the transgene. Colonies were then tested on PMG plates (also containing adenine and uracil) with or without 15 µM thiamine. After two days of growth at 32°C, *nmt41:spi1+* colonies were similar with and without thiamine, while *nmt41:spi1[Q68L]* and *nmt41:spi1[T23N]* colonies appeared normal on plates with thiamine but formed only very tiny colonies on plates without thiamine. *nmt41: spi1+/[Q68L]/[T23N]* overexpression strains were named KS8578, KS8581 and KS8580, respectively.

## Internal deletion of nup146 FG repeats

Strains with internal deletions of *nup146* FG repeats were constructed by a two-step approach (*Fennessy et al., 2014*). For the first step, an *rpl42:natMX6* cassette was amplified by PCR using primer pair OKS2460/OKS2461 and the PCR product was used to transform cycloheximide-resistant *rpl42.sP56Q* strain KS8072. The amplified cassette was at the end of the *nup146* coding sequence to generate a nourseothricin-resistant, cycloheximide-sensitive *nup146:rpl42:natMX6 rpl42.sP56Q* strain, which was named KS8254.

For the second step, a 5.1 kb wild-type *nup146* genomic DNA fragment (containing 5' and 3' untranslated regions as well as coding sequence) was amplified by PCR using primer pair OKS3063/OKS3067. The PCR product was ligated into pJET1.2 vector, and the resulting pJET-*nup146* plasmid was sequenced and named pKS1511. Internal deletions of FG repeats were made within pKS1511 by PCR, using primer pair OKS3093/OKS3094 to make *nup146 [ΔFG5-12Δ]*. The PCR product was recircularized using T4 polynucleotide kinase and T4 DNA ligase, and after transformation, the resulting plasmid was confirmed by sequencing. The pJET-*nup146[ΔFG5-12]* genomic DNA plasmid was named pKS1514.

DNA sequence of *nup146 [ΔFG5-12]* was amplified from pKS1514 by PCR using primer pair OKS3098/OKS3099. The resulting PCR product was transformed into strain KS8254. Nourseothricin-sensitive, cycloheximide-resistant colonies were selected, and colony PCR using primer pair OKS3154/OKS3155 was used to identify the desired strains. The correct *nup146[ΔFG5-12] rpl42.sP56Q* strains was named KS8305.

## Light microscopy

### General light microscopy conditions

Unless stated otherwise, for live-cell microscopy, cells were grown in PMG medium supplemented with adenine, leucine and uracil, with glucose added after autoclaving. Before imaging, cells were grown for 2 days at 25°C, with appropriate dilution to maintain exponential growth. To prepare cells for imaging, 0.5–1 mL of cell culture was centrifuged at 13,000 rpm for 30 s to pellet cells, and a small amount of cell pellet was placed on a pad of 2% agarose in PMG medium supplemented with adenine, leucine and uracil, on a microscope slide. The preparation was then sealed with a coverslip and VALAP (Vaseline, lanolin, and paraffin wax in a 1:1:1 ratio). Preparations were used for ~10–40 min before being discarded.

All microscopy experiments were performed on a spinning-disk confocal microscope, using a Nikon 100x/1.45 NA Plan Apo objective and a Nikon TE2000 inverted microscope (Nikon, Kingston upon Thames, UK) in a 25°C temperature-controlled box, attached to a Yokogawa CSU-10 spinning disk confocal unit (Visitech, Sunderland, UK) and an iXon + Du888 EMCCD camera (Andor, Belfast, UK).

Images were acquired with a step size 0.6 µm and 11 Z-sections for the full cell volume, except for imaging of GFP-tubulin, which used 7 Z-sections. Microscopy images were processed and analyzed by Metamorph (Molecular Devices, San Jose, CA) and Image J software. Figures were prepared using Photoshop and Illustrator (Adobe, San Jose, CA). Graphs and statistical analysis were prepared using Prism (Graphpad, La Jolla, CA). Only linear contrast enhancement was used. Unless otherwise indicated, images are presented as maximum projection of all 11 Z-sections. Images within the same panel in a given figure were all acquired and processed under identical conditions, and therefore signal intensities can be compared directly.

Unless stated otherwise, light-microscopy imaging experiments involved at least two independent biological replicates. We define a biological replicate as growing a fresh culture of a particular strain and taking it through to the end of the experiment. In a few cases, cells were imaged once during strain construction and then once more formally against the appropriate control strain (which itself may have been imaged multiple times). Imaging for *Figure 1—figure supplement 1B* was done once, as this replicates previous work (*Lynch et al., 2014*) and is for illustration purposes only.

### Nuclear positioning

For measuring nuclear position, 0.5 mL of exponentially growing cells was centrifuged at 5000 rpm for 30 s, washed in deionized water, and resuspended in deionized water to a final volume of 10 µL. Cells were fixed by heat-denaturation on a coverslip on a 65°C hot block, and 2.5 µl of mounting media containing DAPI stain was added to fixed cells. Fluorescence images of stained nuclei were acquired together with bright-field images to show the entire cell. For analysis, fluorescence and bright-field images were overlaid using ImageJ, and the distance from the center of the nucleus to the nearest cell tip (S) and the furthest cell tip (L) was measured. We report eccentricity of nuclear position as the ratio S/L (i.e. for a perfectly-centered nucleus, S/L = 1). This experiment was done once, measuring S/L for 100 cells for each genotype.

### Leptomycin B (LMB) treatment

For imaging after LMB treatment, LMB (from 10 µM stock in ethanol) was added to cultures at 100 nM final concentration. For negative controls, ethanol alone was added to an equivalent final concentration (1% v/v). After incubation with or without LMB at 25°C, cells were mounted on medium-agarose pads containing 100 nM LMB and imaged as described above. In *Figure 3—figure supplement 1A* (Alp7-3GFP ± LMB), imaging was done once with 400 nM LMB and once with 100 nM LMB. In *Figure 4F* (GFP-Mto1[1-29]-GST + LMB), imaging was done once. In *Figure 3—figure supplement 3* (MT regrowth), fixation, processing, and imaging was done once for each time-point,

using 200 nM LMB. Because each different time-point was derived from a separate flask, similarities between neighboring time-points indicate reproducibility.

## Spi1 expression

For imaging after expression of *nmt41:spi1+*, *nmt41:spi1[Q68L]*, and *nmt41:spi1[T23N]*, cells were first grown to exponential phase in PMG medium containing adenine, leucine and uracil, plus 15 µM thiamine to repress *nmt41:spi1* expression. Cells of each strain, as well as control cells lacking any *spi1* transgene, were centrifuged at 4000 rpm for 4 min, washed three times with deionized water, resuspended in medium without thiamine, and grown at 25°C until imaging. Preliminary experiments indicated that at this temperature, expression was first noticeable ~26 hr after washing, and more significant at 30–34 hr after washing. Cells were therefore imaged after 26 hr and 34 hr incubation at 25°C.

## Microtubule re-growth after cold-shock

For microtubule regrowth experiments, cells were grown in YE5S liquid medium at 25°C. Manipulations before and after cold-shock, including methanol fixation and processing for immunofluorescence, were performed exactly as described previously (*Lynch et al., 2014*). To assay regrowth after cold-shock, chilled cells were fixed at 34 s, 40 s, 55 s, 70 s, 3 min, and 10 min after being returned to pre-warmed flasks in a 25°C water bath. Cells were stained with TAT1 mouse monoclonal anti-tubulin antibody (1:15 dilution of hybridoma culture supernatant; [*Woods et al., 1989*]) and Alexa488 Donkey anti-mouse secondary antibody (1:80 dilution; Thermo Fisher Scientific). Centrifugation of stained cells onto coverslips for imaging was as described (*Sawin and Nurse, 1998*). This experiment was performed once for each time-point. Because each different time-point was derived from a separate flask, similarities between neighboring time-points indicate reproducibility.

## GFP-tubulin live-cell imaging

For GFP-tubulin live-cell imaging of wild-type (*nup146+*) and *nup146[ΔFG5-12]* cells, cells were imaged every 5 s for a total time of 100 s. Each culture was grown once. Quantification of MT nucleation in the vicinity of the cell nucleus and away from the cell nucleus was determined manually from videos. A total of 90 cells were scored for each of the two genotypes, and total nucleation events were pooled for each genotype.

## Immunoelectron microscopy

Immunoelectron microscopy was carried out as described previously (*Tange et al., 2016*), with some modifications. Briefly, strain KS5750 (*mto1[9A1-NE]-GFP*) was cultured in EMM2 medium with supplements. After washing with 0.1 M phosphate buffer (PB, pH7.4), cells were fixed for 20 min at room temperature with 4% formaldehyde and 0.01% glutaraldehyde dissolved in PB, and washed with PB three times for 5 min each. Cells were then treated with 0.5 mg/mL Zymolyase 100T (Seikagaku Co., Tokyo, Japan) in PB for 30 min. Because the cell walls were not removed well, the cells were further treated with 1 mg/mL Zymolyase 100T in PB for 30 min at 30°C, with 0.2 mg/mL Lysing Enzyme for 30 min, and washed with PB three times. After treatment with 100 mM lysine HCl in PB twice for 10 min and subsequent washing with PB, cells were permeabilized for 15 min with PB containing 0.2% saponin and 1% bovine serum albumin (BSA), and incubated at 4°C overnight with primary antibody (rabbit polyclonal anti-GFP antibody; Rockland, Limerick, PA) diluted 1:400 in PB containing 1% BSA and 0.01% saponin. After washing with PB containing 0.005% saponin three times for 10 min each, cells were incubated for 2 hr at room temperature with secondary antibody (goat anti-rabbit Alexa 594 FluoroNanogold Fab' fragment, Nanoprobes, Yaphank, NY) diluted 1:400 in PB containing 1% BSA and 0.01% saponin, washed with PB containing 0.005% saponin three times for 10 min each, and with PB once. Then, the cells were fixed again with 1% glutaraldehyde in PB for 1 hr, washed with PB once and treated with 100 mM lysine HCl in PB twice for 10 min each. The cells were stored at 4°C until further use.

Before use, the cells were incubated with 50 mM HEPES (pH5.8) three times for 3 min each, washed with distilled water (dH2O) once, and then incubated at 25°C for 3 min with the Silver enhancement reagent (an equal-volume mixture of the following solutions A, B and C: A. 0.2% silver acetate solution. B. 2.8% trisodium citrate-2H$_2$O, 3% citric acid-H$_2$O, and 0.5% hydroquinone. C. 300

mM HEPES, pH 8.2). Cells were then washed with dH$_2$O three times. Cells were embedded in 2% low melting agarose dissolved in dH$_2$O. Then, cells were post-fixed with 2% OsO$_4$ in dH$_2$O for 15 min at room temperature, washed with dH$_2$O three times, stained with 1% uranyl acetate in dH$_2$O for 1 hr, and washed with dH$_2$O three times.

Cells were dehydrated by sequential incubation in 50% and 100% ethanol for 10 min each, and with acetone for 10 min. For embedding in epoxy resin, cells were incubated sequentially with mixtures of acetone: Epon812 (1:1) for 1 hr, acetone:Epon812 (1:2) for 1 hr, and Epon812 overnight, and then Epon812 again for another 3 hr, and left to stand until solidified. The block containing cells was sectioned with a microtome (Leica Microsystems, Tokyo, Japan), and the ultra-thin sections were doubly stained with 4% uranyl acetate for 20 min and lead citrate (Sigma, Tokyo, Japan) for 1 min as usually treated in EM methods. Images were obtained using a JEM1400 transmission electron microscope (JEOL, Tokyo, Japan) at 120kV. Images shown are taken from one of three independent biological replicate experiments.

## Biochemistry and mass spectrometry

### Cell harvesting and cryogrinding

Cell cultures in late exponential growth were collected by centrifugation at 5000 rpm for 15 min at 4°C in a JLA-8.1000 rotor (Beckman Coulter, High Wycombe, UK). Cell pellets were resuspended in one-quarter culture volume of wash buffer (10 mM NaPO$_4$ pH 7.5 and 0.5 mM EDTA) and then washed three times by centrifugation at 5000 rpm for 15 min at 4°C in a JLA-10.500 rotor (Beckman Coulter) and resuspension in the same volume of wash buffer. After the final centrifugation, the cell pellet was weighed and resuspended in wash buffer, using a ratio of 0.3 mL wash buffer per gram of cell pellet. The cell suspension was then quick-frozen by drop-wise addition into liquid nitrogen and stored at −80°C until further use.

Cryogrinding was performed using an RM100 electric mortar grinder with a zirconium oxide mortar and pestle (Retsch, Hope Valley, UK). The mortar and pestle were pre-cooled by filling with liquid nitrogen for 10 min before grinding. Frozen cells were then added into the pre-cooled grinder and ground for 40 min, with regular generous addition of liquid nitrogen to maintain the temperature and prevent cell clumping during the grinding process. Cryogrindate cell powder was recovered and stored at −80°C until further use.

### Anti-GFP immunoprecipitation (for preliminary SILAC interactome analysis)

Large-scale anti-GFP immunoprecipitation was performed using homemade sheep anti-GFP antibody covalently coupled to Protein G Dynabeads (Thermo Fisher Scientific) using dimethylpimelimidate (*Borek et al., 2015*). Immunoprecipitation (IP) buffer contained 15 mM NaPO$_4$ pH 7.5, 100 mM KCl, 0.5 mM EDTA, 0.2% TX-100, 10 μg/mL CLAAPE protease inhibitors (chymostatin, leupeptin, antipain dihydrochloride, aprotinin, pepstatin, E-64), 2 mM AEBSF, 2 mM PMSF, 1 mM NaF, 50 nM calyculin A, 50 nM okadaic acid, 0.1 mM Na$_3$VO$_4$, and 2 mM benzamidine.

After SILAC labeling, cell harvesting and cryogrinding, 37 g of cell cryogrindate powder was mixed with 66.6 mL of cold (4°C) IP buffer and vortexed until dissolved. Cell lysate was then centrifuged at 13,000 rpm for 15 min at 4°C to remove most of the cell debris. The supernatant was transferred to a fresh tube and centrifuged again at 13,000 rpm for 15 min at 4°C, and the second supernatant was recovered. Protein concentration of clarified lysates was measured by Bradford assay and then normalized by adding appropriate volume of IP buffer as necessary.

For immunoprecipitation, 140 μL of anti-GFP/Protein G-Dynabead slurry (~2.1×10$^9$ beads, coupled to ~85 μg of antibody) were washed twice with 0.5 mL of IP buffer, mixed with 70 mL of clarified cell lysate, and incubated at 4°C for 1.5 hr with gentle rotation. Beads were then collected with a magnet, washed three times with 1 mL IP buffer, transferred to a fresh microfuge tube, and washed twice again with 1 mL IP buffer. The beads were then centrifuged at 13,000 rpm for 10 s, and any remaining buffer was removed.

To elute proteins from beads, a total of 65 μL Laemmli sample buffer (LSB; 2% SDS (v/v), 10% glycerol, 62.5 mM Tris pH 6.8) was added to beads, which were then mixed by pipetting and incubated at 65°C for 15 min with intermittent vortexing. The mixture was then briefly centrifuged before transferring the supernatant to a fresh microfuge tube, and DTT and bromophenol blue were added

to final concentrations of 0.1 M and 0.01%, respectively. This final sample was then heated at 95°C for 5 min and stored at −20°C prior to SDS-PAGE.

Samples from large-scale immunoprecipitations were processed for SILAC mass spectrometry analysis as described below.

## Cryogrindate cross-linking in vitro

A 0.125 M stock solution of disuccinimidyl suberate (DSS; Thermo Fisher Scientific) was made fresh in DMSO. Just before cross-linking, this was diluted to 2.5 mM final concentration in cross-linking buffer (15 mM NaPO4 pH 7.5, 85 mM NaCl, 0.2% Triton X-100, 1 mM PMSF, 10 µg/mL CLAAPE protease inhibitors, 2 mM AEBSF, 1 mM NaF, 50 nM okadaic acid, 0.1 mM $Na_3VO_4$ and 2 mM benzamidine). Of cell cryogrindate powder, 6 g was resuspended in 6 mL of cross-linking buffer containing DSS and mixed by vortexing. Cell lysate was then incubated at 4°C for 2 hr with gentle rotation. Then, 1.2 mL of 1.5 M Tris-HCl pH 8.8 was added to the cell lysate to quench the cross-linking reaction, and left at room temperature for 30 min. The cross-linked cell lysate was then used for two-step purification as described below.

## Two-step purification of HTB-tagged Mto1 variants

HTB-tagged Mto1 variants were purified in two steps, using nickel-charged Fractogel EMD Chelate (M) (Merck, Darmstadt, Germany) resin and Nanolink magnetic streptavidin beads (Trilink, San Diego, CA), under denaturing conditions. The procedure described below was used for purifications after cryogrindate cross-linking in vitro, which was most commonly used. For purifications after cross-linking in vivo, the same approach was used, but all amounts and volumes were doubled (this is because initial purifications were done after cross-linking in vivo, and it was later determined that half as much material was still sufficient for MS analysis)

For the first-step purification, 12 mL of cross-linked and quenched cell lysate (representing 6 g cryogrindate) was mixed with 60 mL guanidine purification buffer (6 M guanidine, 15 mM $NaPO_4$ pH 7.5, 85 mM NaCl, 0.5% TritonX-100, 1 mM PMSF, 1 mM NaF, 0.1 mM $Na_3VO_4$ and 2 mM benzamidine). The cell lysate was then sonicated with a Sonics VC505 sonicator fitted with a 3 mm tip for 2 min (1 s on, 1 s off, for total time 4 min, at 60% amplification), centrifuged at 4000 rpm for 15 min at room temperature to remove cell debris, and the supernatant was recovered. 1.2 mL of 50% slurry of Fractogel EMD Chelate (henceforth referred to as 'Fractogel') was charged with nickel and washed twice with 5 mL distilled water, and twice with 5 mL guanidine purification buffer. The charged Fractogel bed was resuspended with an equal volume of 6 M guanidine purification buffer and mixed with the lysate supernatant and incubated at room temperature for 2 hr, with gentle rotation. The suspension was then transferred to a 20 mL disposable plastic column (Evergreen Scientific, Rancho Dominguez, CA), washed once with 20 mL of 6 M guanidine purification buffer, and washed 3 times with 20 mL of 8M urea purification buffer (contained 8 M urea, 15 mM $NaPO_4$ pH 7.5, 85 mM NaCl, 0.1% TritonX-100, 1 mM PMSF, 1 mM NaF, 0.1 mM $Na_3VO_4$ and 2 mM benzamidine). The Fractogel was then resuspended in 1 mL of 8 M urea purification buffer and transferred into a 15 mL polypropylene tube. This process was repeated for two more times to recover all of the Fractogel from the column. Fractogel was then centrifuged at 4000 rpm for 3 min at RT. The supernatant was discarded, and 3 mL of LSB containing 600 mM imidazole was added to the tube, and bound proteins were eluted by heating at 95°C for 5 min. The Fractogel was then centrifuged at 4000 rpm for 3 min, and the supernatant was recovered, quick-frozen in liquid nitrogen, and stored at −80°C.

For the second-step purification, the stored elution from the first-step purification above was thawed at room temperature, and TX-100 was added to a final concentration of 1%. 30 µL of Nanolink streptavidin beads slurry (as supplied by manufacturer; this corresponds to ~1.2 µL bed volume) was washed twice with 1 mL of LSB containing 1% TX-100, resuspended into 30 µL of LSB containing 1% TX-100 and added to the thawed first-step elution. This suspension was incubated for 1.5 hr at room temperature, then collected with a magnet and washed once with 1 mL of LSB and three times with 1 mL of LSB without glycerol. After transfer to a microfuge tube, beads were resuspended in 15 µL of LSB and heated at 95°C for 5 min. The elution from the beads was collected and DTT and bromophenol blue were added, to final concentrations of 0.1M and 0.01%, respectively. The mixture was boiled again for 5 min and stored at −20°C prior to SDS-PAGE.

## Mass spectrometry (label-free quantification)

For label-free quantification mass spectrometry analysis of samples after two-step purification,~18 µL of second-step elution was loaded onto a single lane (~0.5 cm wide) of a 4–20% Tris-glycine poly-acrylamide gel (Biorad, Watford, UK). Samples were run at 150V for 12–14 min. The gel was stained with Coomassie Blue at room temperature for 1 hr and destained in 10% acetic acid overnight. On the following day, the gel was washed once in distilled water and the relevant region recovered after excision with a clean scalpel. In general, for all samples, we recovered the region of the gel above, but not including, the non-cross-linked Mto1 band, in order to increase the relative proportion of cross-linked Mto1 species vs. non-crosslinked Mto1.

Excised gel bands were destained with 50 mM ammonium bicarbonate (Sigma Aldrich, Gillingham, UK) and 100% (v/v) acetonitrile (Sigma Aldrich) and proteins were digested with trypsin, as previously described (*Shevchenko et al., 1996*). In brief, proteins were reduced in 10 mM dithio-threitol (Sigma Aldrich) for 30 min at 37°C and alkylated in 55 mM iodoacetamide (Sigma Aldrich) for 20 min at ambient temperature in the dark. They were then digested overnight at 37°C with 13 ng/µL trypsin (Thermo Fisher Scientific).

Following digestion, samples were diluted with an equal volume of 0.1% TFA and spun onto StageTips as described (*Rappsilber et al., 2003*). Peptides were eluted in 40 µL of 80% acetonitrile in 0.1% TFA and concentrated to 1 µL by vacuum centrifugation (Concentrator 5301, Eppendorf, Stevenage, UK). Samples were then prepared for LC-MS/MS analysis by diluting to 5 µL with 0.1% TFA. LC-MS-analyses were performed on a Q Exactive mass spectrometer (Thermo Fisher Scientific) (*Figures 1*, *3* and *6*) and on an Orbitrap Fusion Lumos Tribrid Mass Spectrometer (Thermo Fisher Scientific) (*Figure 4*), both coupled on-line to Dionex Ultimate 3000 RSLCnano Systems (Thermo Fisher Scientific). Peptides were separated on a 50 cm EASY-Spray column (Thermo Fisher Scientific) assembled in an EASY-Spray source (Thermo Fisher Scientific) and operated at 50°C. In both cases, mobile phase A consisted of 0.1% formic acid in water while mobile phase B consisted of 80% aceto-nitrile and 0.1% formic acid. Peptides were loaded onto the column at a flow rate of 0.3 µL/min and eluted at a flow rate of 0.2 µL/min according to the following gradients: 2% to 40% buffer B in 90 min, then to 95% buffer B in 11 min (*Figures 1*, *3* and *4*) and 2% to 40% buffer B in 120 min and then to 95% buffer B in 11 min (*Figure 6*). For Q Exactive, FTMS spectra were recorded at 70,000 resolution (scan range 350–1400 m/z) and the ten most intense peaks with charge ≥2 of the MS scan were selected with an isolation window of 2.0 Thomson for MS2 (filling 1.0E6 ions for MS scan, 5.0E4 ions for MS2, maximum fill time 60 ms, dynamic exclusion for 50 s). For Orbitrap Fusion Lumos, survey scans were performed at 60,000 resolution (scan range 350–1400 m/z) with an ion tar-get of 7.0e5. MS2 was performed in the orbitrap with ion target of 5.0E3 and HCD fragmentation with normalized collision energy of 25 (*Olsen et al., 2007*). The isolation window in the quadrupole was 1.6. Only ions with charge between 2 and 7 were selected for MS2.

The MaxQuant software platform (*Cox and Mann, 2008*) version 1.5.2.8 was used to process raw files, and search was conducted against *Schizosaccharomyces pombe* complete/reference proteome set from PomBase (www.pombase.org; released in July, 2016; *McDowall et al., 2015*), using the Andromeda search engine (*Cox et al., 2011*). The first search peptide tolerance was set to 20 ppm while the main search peptide tolerance was set to 4.5 ppm. Isotope mass tolerance was set to 2 ppm, and maximum charge state was set to 7. Maximum of two missed cleavages were allowed. Carbamidomethylation of cysteine was set as fixed modification. Oxidation of methionine and acety-lation of the N-terminal were set as variable modifications. Label-free quantification analysis was per-formed by employing the MaxLFQ algorithm as described (*Cox et al., 2014*). Peptide and protein identifications were filtered to 1% FDR.

All LFQ MS analyses were performed using two complete biological replicates of each of the two conditions being compared. For experiments shown in *Figure 6*, additional biological replicates using cells grown in LowN medium were also performed, alongside those using cells grown in PMG (as normal). Including the LowN replicates during MaxQuant analysis improved the quality of peptide identifications from experiments using cells grown in PMG. Data shown in *Figure 6* are only from cells grown in PMG; however, data from cells grown in LowN are completely consistent with the data from cells grown in PMG and are included with the PMG data as part of *Supplementary file 6*.

Scatterplots showing LFQ ratio vs. LFQ intensity (*Figures 1*, *3*, *4* and *6*) were constructed as fol-lows: In cases where the relevant Mto1-interactors (e.g. Crm1, Nup146) were fully quantified in both

conditions of both replicate experiments (i.e. *Figure 3 and 6*), the values shown in scatterplots represent the geometric mean from the two replicates. The geometric mean was used rather than the arithmetic mean in order to minimize any effects of extreme outliers. In other cases (i.e. *Figure 1 and 4*), Nup146 was not fully quantified in one of the conditions of one of the replicate experiments, because of low signal intensity or low peptide count. In these cases, it was not possible to calculate mean LFQ values for Nup146 from replicate experiments, and therefore, the values shown in scatterplots are taken from the replicate in which the Nup146 was fully quantified. *Tables 1–4* show peptide counts and LFQ values for selected proteins from replicate experiments, and *Supplementary file 3–6* contain full datasets, including LFQ values.

## Mass spectrometry (SILAC)

To generate the SILAC data shown in *Supplementary file 2* (preliminary results, from anti-GFP immunoprecipitation), sample processing and digestion was performed as described above. LC-MS analyses were performed on a Q Exactive mass spectrometer (Thermo Fisher Scientific) coupled on-line to a Dionex Ultimate 3000 RSLCnano System (Thermo Fisher Scientific). The analytical column with a self-assembled particle frit (*Ishihama et al., 2002*) and C18 material (ReproSil-Pur C18-AQ 3 µm; Dr. Maisch GmbH, Ammerbuch, Germany) was packed into a spray emitter (75 µm ID, 8 µm opening, 300 mm length; New Objective) using a Proxeon air-pressure pump (Thermo Fisher Scientific). Mobile phase A consisted of 0.1% formic acid in water while mobile phase B consisted of 80% acetonitrile and 0.1% formic acid. Peptides were loaded onto the column at a flow rate of 0.5 µL/min and eluted at a flow rate of 0.2 µL/min according to the following gradient: 2% to 40% in 120 min and then to 95% in 11 min. The settings on the Q Exactive were the same as described above.

The MaxQuant software platform (*Cox and Mann, 2008*) version 1.3.0.5 was used to process raw files, and search was conducted against *Schizosaccharomyces pombe* complete/reference proteome set from PomBase (released in August, 2012), using the Andromeda search engine (*Cox et al., 2011*). The first search peptide tolerance was set to 20 ppm, while the main search peptide tolerance was set to 4.5 ppm. Isotope mass tolerance was two ppm and maximum charge was set to 7. The MS/MS match tolerance was set to 20 ppm, and two missed cleavages were allowed. Carbamidomethylation of cysteine was set as fixed modification, and oxidation of methionine with acetylation of the N-terminal were set as variable modifications. Multiplicity was set to 2, and for heavy labels, Arginine-6 and Lysine-8 were selected, and peptide and protein identifications were filtered to 1% FDR. Unique and non-unique peptides were used for quantification. Proteins with minimum of two quantified labeled peptide pairs/triplets were reported for quantification, and the isoforms with the highest peptide counts were considered for quantification.

The mass spectrometry proteomics data from both LFQ and SILAC experiments have been deposited to the ProteomeXchange Consortium via the PRIDE (*Vizcaíno et al., 2016*) partner repository with the dataset identifier PXD008334.

## Acknowledgements

We thank I Hagan, S Oliferenko, M Sato, and K Weis for yeast strains, plasmids and/or reagents. We thank members of our labs for helpful discussions, and A Cook for insights on nuclear transport and for comments on the manuscript. This work was supported by the Wellcome Trust ([094517] to KES, and [108504], [091020] to JR), and by KAKENHI grants from the Japan Society for the Promotion of Science (JP25116006 and JP17H03636 to TH, and JP17H01444 and JP16H01309 to YH). XXB was also supported by the Darwin Trust of Edinburgh. The Wellcome Centre for Cell Biology is supported by core funding from the Wellcome Trust [203149].

## Additional information

### Funding

| Funder | Grant reference number | Author |
|---|---|---|
| Wellcome | 094517 | Xun X Bao<br>Eric M Lynch<br>Kenneth E Sawin |
| Japan Society for the Promotion of Science | JP25116006 | Tomoko Kojidani<br>Tokuko Haraguchi |
| The Darwin Trust of Edinburgh | | Xun X Bao |
| Wellcome | 108504 | Christos Spanos<br>Juri Rappsilber |
| Wellcome | 091020 | Christos Spanos<br>Juri Rappsilber |
| Wellcome | 203149 | Xun X Bao<br>Christos Spanos<br>Eric M Lynch<br>Juri Rappsilber<br>Kenneth E Sawin |
| Japan Society for the Promotion of Science | JP17H03636 | Tomoko Kojidani<br>Tokuko Haraguchi |
| Japan Society for the Promotion of Science | JP17H01444 | Yasushi Hiraoka |
| Japan Society for the Promotion of Science | JP16H01309 | Yasushi Hiraoka |

The funders had no role in study design, data collection and interpretation, or the decision to submit the work for publication.

### Author contributions

Xun X Bao, Christos Spanos, Formal analysis, Investigation, Writing—original draft, Writing—review and editing; Tomoko Kojidani, Investigation, Writing—review and editing; Eric M Lynch, Formal analysis, Investigation, Writing—review and editing; Juri Rappsilber, Yasushi Hiraoka, Tokuko Haraguchi, Supervision, Writing—review and editing; Kenneth E Sawin, Conceptualization, Formal analysis, Supervision, Writing—original draft, Writing—review and editing

### Author ORCIDs

Xun X Bao http://orcid.org/0000-0003-4733-3550
Christos Spanos http://orcid.org/0000-0002-4376-8242
Eric M Lynch https://orcid.org/0000-0001-5897-5167
Juri Rappsilber http://orcid.org/0000-0001-5999-1310
Yasushi Hiraoka http://orcid.org/0000-0001-9407-8228
Tokuko Haraguchi http://orcid.org/0000-0002-3813-6785
Kenneth E Sawin http://orcid.org/0000-0002-2607-2219

### Decision letter and Author response

Decision letter https://doi.org/10.7554/eLife.33465.034
Author response https://doi.org/10.7554/eLife.33465.035

## Additional files

### Supplementary files

• Supplementary file 1. Yeast strains, plasmids and oligonucleotides used in this work.
DOI: https://doi.org/10.7554/eLife.33465.024
• Supplementary file 2. Mass spectrometry data and summary from preliminary SILAC experiment.

DOI: https://doi.org/10.7554/eLife.33465.025

• Supplementary file 3. Combined mass spectrometry data and LFQ summaries for experiments shown in *Figure 1*.
DOI: https://doi.org/10.7554/eLife.33465.026

• Supplementary file 4. Combined mass spectrometry data and LFQ summaries for experiments shown in *Figure 3*.
DOI: https://doi.org/10.7554/eLife.33465.027

• Supplementary file 5. Combined mass spectrometry data and LFQ summaries for experiments shown in *Figure 4*.
DOI: https://doi.org/10.7554/eLife.33465.028

• Supplementary file 6. Combined mass spectrometry data and LFQ summaries for experiments shown in *Figure 6*.
DOI: https://doi.org/10.7554/eLife.33465.029

• Transparent reporting form
DOI: https://doi.org/10.7554/eLife.33465.030

### Data availability

The mass spectrometry proteomics data from both LFQ and SILAC experiments have been deposited to the ProteomeXchange Consortium via the PRIDE partner repository with the dataset identifier PXD008334.

The following dataset was generated:

| Author(s) | Year | Dataset title | Dataset URL | Database, license, and accessibility information |
|---|---|---|---|---|
| Bao XX, Spanos C, Kojidani T, Lynch EM, Rappsilber J, Hiraoka Y, Haraguchi T, Sawin KE | 2017 | Exportin Crm1 is repurposed as a docking protein to generate microtubule organizing centers at the nuclear pore | https://www.ebi.ac.uk/pride/archive/projects/PXD008334 | Publicly available at EBI PRIDE (accession no. PXD008334) |

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
