## [Decision Letter]

Thank you for submitting your article "Exportin Crm1 is repurposed as a docking protein to generate microtubule organizing centers at the nuclear pore" for consideration by *eLife*. Your article has been reviewed by two peer reviewers, and the evaluation has been overseen by a Reviewing Editor and Andrea Musacchio as the Senior Editor. The reviewers have opted to remain anonymous.

The reviewers have discussed the reviews with one another and the Reviewing Editor has drafted this decision to help you prepare a revised submission.

Summary:

Microtubules (MTs) nucleate from the nuclear envelope (NE) in many cell types including fission yeast, but how the MT-nucleating activity is retained at the NE remains unknown. In this interesting manuscript, the authors use comparative mass spectrometry to identify interactors of Mto1, a receptor for γ-tubulin complexes. They describe a novel mechanism of microtubule organization, involving recruitment of Mto1 to the cytoplasmic face of nuclear pores via the nuclear export factor Crm1 and the nucleoporin Nup146. The reviewers agreed that the experiments are overall well executed and that this manuscript is a good fit for *eLife*. They have a few requests for revisions:

Essential revisions:

1) The authors' model implies that there should be a fairly stable Crm1 pool at the sites of MT attachment to the NE in wild type cells. Is this the case? The authors could use FLIP or similar approaches with Crm1-GFP to detect sites with high Crm1 residency times.

2) Does short LMB treatment lead to disappearance of nucleus-nucleated/anchored MTs in wild type cells?

3) The authors propose that the Crm1 is bound to RanGTP when it anchors Mto1 at the NPCs. This model appears to be based solely on their interpretation of RanT24N overexpression that causes displacement of Mto1-NE protein from the NE. Yet, RanT24N should also block Crm1-dependent nuclear export potentially exhausting the Crm1 pool in the cytoplasm. Is it possible to provide direct evidence for the presence of Ran in Crm1-Mto1-Nup146 complexes? If this is not possible within the 2 months revision time the authors should at least discuss alternative possibilities.

4) It would be of interest to the readership to discuss whether the described mechanisms have any resemblance to Nup-dependent γ-tubulin recruitment at the kinetochores, for the nucleation of spindle microtubules, since Crm1 has also been localized there (Arnaoutov et al., 2005; Mishra et al., 2010).

---

## [Author Response]

Essential revisions:

1) The authors' model implies that there should be a fairly stable Crm1 pool at the sites of MT attachment to the NE in wild type cells. Is this the case? The authors could use FLIP or similar approaches with Crm1-GFP to detect sites with high Crm1 residency times.

We agree that our model implies that Mto1 docking should in principle lead to some increase in the amount and/or stability of Crm1 at the NE. However, for technical reasons, the proposed experiment (or a similar approach, such as measuring possible changes in Crm1 levels at the NE as a result of Mto1 docking) does not appear to be currently achievable.

The proposed experiment(s) requires a fully functional, fluorescent-tagged version of Crm1. We constructed a fluorescent-tagged version of Crm1 at the endogenous *crm1* locus, and cells were viable, suggesting that nuclear export was not significantly adversely affected by the tagging. However, we unexpectedly found that the fluorescent-tagged Crm1 abrogated Mto1 localization to the NE. A 13Myc-tagged version of Crm1 also abrogated Mto1 localization to the NE, suggesting that the problem is related to Crm1 tagging generally rather than fluorescent-tagging per se.These findings make it currently impossible to perform an experiment of the type suggested.

A related point is that even if Crm1 tagging did not affect Mto1 localization to the NE, it is not clear whether it would be possible to detect significant quantitative changes in Crm1 localization and/or dynamics as a consequence of Mto1 docking. This is because Crm1 is almost certainly much more abundant in cells than Mto1 (or Mto2). In work separate from the current manuscript, we measured Mto1 and Mto2 levels in vivo and found that each is present at about ~2700 molecules per cell. By contrast, proteomics data available on PomBase suggests that Crm1 is maybe 10X more abundant, and possibly more. This means that at best, only a very small fraction of the total Crm1 pool could be affected by Mto1 docking; this might be below the fluorescence “noise level” for live-cell experiments.

Overall, while we agree that this could have been interesting experiment, we would also argue that it is not absolutely critical to our model, which is supported by many other experiments.

2) Does short LMB treatment lead to disappearance of nucleus-nucleated/anchored MTs in wild type cells?

Yes, it does. In the revised manuscript, we show this in the new Figure 3—figure supplement 3. We treated cells with LMB and then assayed microtubule regrowth after cold-induced depolymerization (the experiment is of the same general design as in Figure 6). As predicted, in LMB-treated cells, microtubule regrowth does not occur from the nuclear envelope and instead occurs randomly in the cytoplasm.

3) The authors propose that the Crm1 is bound to RanGTP when it anchors Mto1 at the NPCs. This model appears to be based solely on their interpretation of RanT24N overexpression that causes displacement of Mto1-NE protein from the NE. Yet, RanT24N should also block Crm1-dependent nuclear export potentially exhausting the Crm1 pool in the cytoplasm. Is it possible to provide direct evidence for the presence of Ran in Crm1-Mto1-Nup146 complexes? If this is not possible within the 2 months revision time the authors should at least discuss alternative possibilities.

Following the reviewers’ suggestion, in the revised Discussion (subsection “A novel function for cytoplasmic FG-Nups?”) we now explicitly discuss the possibility that Ran may not be present in the Crm1-Mto1-Nup146 complexes, and that the requirement for RanGTP for Mto1 docking at NPCs could be indirect. We have not changed the figures, because we think the analogy with known mechanisms of conventional cargo export is still quite reasonable.

In order to have a more balanced Discussion, we also now point out that our cross-linking MS experiments are inconclusive as to whether Ran is present in Crm1-Mto1-Nup146 complexes. On the one hand, the LFQ MS data in Supplementary files 3-6 do not indicate any enrichment of Ran with Mto1 when Mto1 is localized to the NE, compared to when Mto1 is not localized to the NE (from the 10 different MS experiments, the mean MaxQuant “enrichment” ratio for Ran is 1.1 +/- 0.20 (SD)). On the other hand, however, it should be noted that in order for Ran to be enriched in these types of experiments, it must be cross-linked to Mto1, either directly or indirectly. If Ran was not easily cross-linked – for whatever reason – we would fail to detect enrichment, even if Ran was present in the Crm1-Mto1-Nup146 complex.

4) It would be of interest to the readership to discuss whether the described mechanisms have any resemblance to Nup-dependent γ-tubulin recruitment at the kinetochores, for the nucleation of spindle microtubules, since Crm1 has also been localized there (Arnaoutov et al., 2005; Mishra et al., 2010).

We have now expanded the scope of the Discussion in this area (fourth paragraph). Our original submission cited Arnaoutov *et al.* but did not discuss it in detail.